# Single-cell isotope tracing reveals functional guilds of bacteria associated with the diatom *Phaeodactylum tricornutum*

Xavier Mayali [1] ✉, Ty J. Samo [1], Jeff A. Kimbrel [1], Megan M. Morris [1], Kristina Rolison [1], Courtney Swink [1], Christina Ramon[1], Young-Mo Kim [2], Nathalie Munoz-Munoz [2], Carrie Nicora [2], Sam Purvine [2], Mary Lipton[2], Rhona K. Stuart [1] & Peter K. Weber [1]

Bacterial remineralization of algal organic matter fuels algal growth but is rarely quantified. Consequently, we cannot currently predict whether some bacterial taxa may provide more remineralized nutrients to algae than others. Here, we quantified bacterial incorporation of algal-derived complex dissolved organic carbon and nitrogen and algal incorporation of remineralized carbon and nitrogen in fifteen bacterial co-cultures growing with the diatom *Phaeodactylum tricornutum* at the single-cell level using isotope tracing and nano-SIMS. We found unexpected strain-to-strain and cell-to-cell variability in net carbon and nitrogen incorporation, including non-ubiquitous complex organic nitrogen utilization and remineralization. We used these data to identify three distinct functional guilds of metabolic interactions, which we termed macromolecule remineralizers, macromolecule users, and small-molecule users, the latter exhibiting efficient growth under low carbon availability. The functional guilds were not linked to phylogeny and could not be elucidated strictly from metabolic capacity as predicted by comparative genomics, highlighting the need for direct activity-based measurements in ecological studies of microbial metabolic interactions.

Algal-bacterial interactions are fundamental to primary productivity in the oceans and other surface waters, including algal bioenergy and bioproduct production ponds[1]. Studies have shown that bacteria can increase algal productivity by providing vitamins[2], siderophores[3], and algal growth hormones[4]. More fundamentally, bacteria can remineralize nutrients, through for example, the deamination of organic nitrogen to ammonium[5]. It is generally assumed that algae-associated bacteria grow on algal-derived organic matter, making these interactions mutualistic or at least commensal (i.e., the bacteria benefit and the algae are unaffected), but few studies have empirically measured the fraction of bacterial carbon (C) and nitrogen (N) that is derived from photosynthetic microalgae. To our knowledge, the reverse process (transfer from bacteria to algae) has never been directly quantified. These microscale exchanges, when integrated over large volumes in outdoor algal cultures and in aquatic ecosystems, have profound consequences for elemental cycling[6].

The last few decades have greatly increased our knowledge of the mechanisms of bacterial C and N recycling in aquatic ecosystems. Algal blooms support a microbial community that progresses over time in response to the changing availability of algal-derived organic matter over the course of weeks[7] and as quickly as within one day[8]. Generally, algal-associated bacteria that recycle algal organic matter have been divided into those that specialize on polymers and macromolecules, dominated by *Bacteroidia*, and those that specialize in smaller

---

[1]Physical and Life Sciences Directorate, Lawrence Livermore National Laboratory, Livermore, CA, USA. [2]Earth and Biological Sciences Directorate, Pacific Northwest National Laboratory, Richland, WA 99352, USA. ✉e-mail: mayali1@llnl.gov

molecules, dominated by *Rhodobacterales*[9]. In both cases, the primary mechanisms for degradation and uptake involve extracellular enzymes and transporters[10], and many resources now exist to identify genes involved in such processes, including polysaccharide[11] and protein[12] degradation. Compounds released by bacteria and known to be reincorporated by photoautotrophs for biomass include ammonium[5], amino acids[13], and carbon dioxide[14]. However, nutrient remineralization has not been identified as a primary mechanism of algal growth promotion by bacteria[15], perhaps because in nature, they are often decoupled in time and space[16]. Nonetheless, since different bacteria are known to consume different sources of organic matter, it seems plausible that different bacteria likewise express different rates of nutrient remineralization. Thus, there is a need to measure net exchanges of C and N between algae and bacteria, identify whether these exchanges are correlated to mutualism or physical association and whether the quantity can be predicted by phylogenetic affiliation or genomic content.

A practical approach to quantify metabolic activity and exchange is the use of stable isotope incubations. DNA-stable isotope probing (SIP) was developed to identify organisms that incorporate a pure substrate, for example, $^{13}$C-labeled methanol[17]. Further studies have used isotope-labeled extracts from phytoplankton to qualitatively identify bacterial taxa that incorporated these mixed substrates[18]. A more quantitative analysis from such incubations can be carried out at the single-cell level using a nanoSIMS imaging secondary ion mass spectrometer through an approach called nanoSIP[19]. This high-resolution method allows the quantification of isotope incorporation by single cells, including small cells that are attached to one another[20]. For example, nanoSIP has been used with mixed bacterial communities growing with one phototrophic alga to show that different bacterial communities have different effects on algal growth and cell-specific carbon fixation, and in some cases this mutualism is mediated by cell-to-cell attachment[21]. However, working with mixed communities makes it challenging to attribute specific impacts to individual taxa, and is also complicated by bacteria-bacteria interactions that might obscure otherwise recognizable interactions. These inherent challenges therefore required us to develop modified culturing and SIP approaches that enable taxon-specific probing and focused observations of bacterial activity and exchange with their algal partner.

Here, we relate the identity, genetic potential, and activity of algal-associated bacteria using isotope tracing to quantify the transfer of complex algal C and N to bacteria and remineralized C and N to the algae. We define 'remineralization' in these experiments by measurements of net C and N incorporated by the algal cells. Thus, this does not represent gross remineralization of algal organic matter, but rather what is assimilated by the algal cells as they are actively growing, prior to nutrients becoming depleted. Linked activity and metabolic potential[22] can be useful in order to model complex microbial processes, for example using functional trait approaches[23]. We isolated fifteen representative bacteria from mixed community enrichments previously growing with the diatom *Phaeodactylum tricornutum* under autotrophic conditions. We reinoculated these bacterial isolates into previously bacteria-free *P. tricornutum*, allowed these co-cultures to reach equilibrium ratios of algae to bacteria based on metabolic exchange, and tested their impact on algal biomass yield. Then we added chemically extracted isotope-labeled *P. tricornutum*-derived organic matter to these co-cultures to measure organic C and N incorporation by the bacterial cells as well as remineralization of C or N back to *P. tricornutum* cells. We further examined bacterial incorporation of nitrate in the presence of glycolate, a simple C source known to be produced by *P. tricornutum*. In addition to comparative genomics analyses of the 15 strains, we focused on two bacterial isolates representing contrasting phenotypes, linking genotype and phenotype by investigating their most abundantly expressed proteins during growth in algal cultures. All together, our data led us to develop

a conceptual framework of bacterial association with microalgae that takes into account nutrient exchange, competition, and metabolism, suggesting that algal-associated bacteria in our experiments can be classified into three major functional classes (or 'guilds') that exhibit distinct strategies allowing them to proliferate in high nutrient and high-density algal systems.

## Results

### Growth impacts, origin, and niche of *P. tricornutum* associated bacteria

Although the culture collection is by no means exhaustive, it spans the major taxonomic groups associated with *P. tricornutum* and provides a practical set of co-cultures that can be tested experimentally to examine patterns of nutrient exchange with their algal host. Numerically dominant taxa associated with *P. tricornutum*, as measured with cultivation independent 16S rRNA gene sequencing, primarily belong to the Gammaproteobacteria, Alphaproteobacteria, and Bacteroidia[24]. Our isolation efforts yielded strains from 15 unique genera, all belonging to these three taxonomic groups (Table 1), and represent 16S rRNA gene variants identified in outdoor *P. tricornutum* cultures and multi-species laboratory enrichments (Fig. S1). The majority of the isolates did not impact *P. tricornutum* in-vivo chlorophyll fluorescence (a measure of algal biomass; Table 1, Fig. S2), with the exception of *Marinobacter*, *Devosia*, and *Alcanivorax*, the presence of which increased fluorescence in early stationary growth phase compared to axenic cultures, as previously reported[25]. Four bacterial strains were detected attached to *P. tricornutum* cells (*Marinobacter*, *Oceanicaulis*, *Roseibium*, and *Yoonia*), all of which originated from phycosphere enrichments where free-living bacteria were washed away[21]. We note that most bacterial cells in these co-cultures were still free-living, with a small percentage detected as algal-attached. We also examined the total cell abundances of the bacterial strains grown in co-culture with no other sources of organic matter besides photosynthetic exudation by *P. tricornutum*. Other than *Henriciella*, which was present at such low abundance that it was lower than background counts in the axenic culture, the bacteria in the co-cultures exhibited abundances ranging from $1 \times 10^3$ to $2 \times 10^7$ cells mL$^{-1}$ (Table 1). There was no statistically significant association between bacterial cell abundances and strain origin (free-living vs. attached, or phycosphere enrichments vs. algal culture enrichments, one-way ANOVA, $p > 0.05$).

### Bacterial incorporation and remineralization of complex algal organic matter

We first aimed to quantify bacterial incorporation of C and N from algal exudates during co-cultivation. To address this, we added solid-phase extracted $^{13}$C- and $^{15}$N-labeled extracellular organic matter from axenic *P. tricornutum* to the 15 algae-bacteria co-cultures and used nanoSIMS to quantify net bacterial C and N incorporation after 24 h. Not including formalin-killed controls, we collected $^{13}$C and $^{15}$N incorporation data for a total of 445 bacterial cells (Fig. 1A, S3, S4), finding high variability of net incorporation of substrate C and N ($C_{net}$ and $N_{net}$) among the isolates, as well as cell-to-cell variability for each isolate. All tested bacteria significantly incorporated algal DON (compared to killed controls) except *Tepidicaulis*, *Marinobacter*, and *Alcanivorax* (Kruskal-Wallis multiple comparison test, p values 0.2, 0.27, and 0.83, respectively; Fig. 1A, S4A). Among the isotopically enriched strains, there was a wide range of incorporation, from median daily $N_{net}$ values of 0.4% to 16%. Strains with >5% of their daily N biomass from added algal DON included *Muricauda*, *Algoriphagus*, *Oceanicaulis*, *Yoonia*, *Pusillimonas*, *Devosia*, and *Arenibacter*. Some of the statistically significantly enriched strains had low daily $N_{net}$ values (<1%), including *Roseibium* and *Thalassospira*. Regarding cell-to-cell variability, we calculated coefficient of dispersion (CD, Table 1), finding a range of 16 to 62%. Notably, the strains with the highest daily $N_{net}$ also exhibited higher cell-to-cell variability, with CD values greater than 30%. We also

**Table 1 | Summary of 15 *P. tricornutum* associated bacteria investigated in this study**

| genus and strain | Genome Taxonomy Database Identification | Genbank accession | isolation source | bacteria cells mL$^{-1}$ | algal attached | higher algal biomass | bacteria Nnet CD | bacteria Cnet CD | μg C from DOC L$^{-1}$ d$^{-1}$ | μg N from DON L$^{-1}$ d$^{-1}$ |
|---|---|---|---|---|---|---|---|---|---|---|
| Oceanicaulis PT13A | Alphaproteobacteria;Caulobacterales;Maricaulaceae | PRJNA441690 | phycosphere | 4.39E+06 | yes | no | 62% | 19% | 12.9 | 6.2 |
| Algoriphagus ARW1R1 | Bacteroidia;Cytophagales;Cyclobacteriaceae | PRJNA441694 | free-living | 1.74E+06 | no | no | 31% | 28% | 6.8 | 3.2 |
| Muricauda ARW1Y1 | Bacteroidia;Flavobacteriales;Flavobacteriaceae | PRJNA581034 | free-living | 3.09E+06 | no | no | 44% | 15% | 10.0 | 8.7 |
| Arenibacter ARW7G5Y1 | Bacteroidia;Flavobacteriales;Flavobacteriaceae; | PRJNA441693 | free-living | 3.67E+06 | no | no | 58% | 37% | 8.3 | 4.2 |
| Devosia PTEAB7WZ | Alphaproteobacteria;Rhizobiales;Devosiaceae | PRJNA441687 | free-living | 1.02E+06 | no | yes | 16% | 31% | 2.7 | 1.2 |
| Stappia ARW1T | Alphaproteobacteria;Rhizobiales;Stappiaceae | PRJNA441689 | free-living | 1.33E+06 | no | no | 20% | 11% | 3.9 | 0.4 |
| Rhodophyticola 6CLA | Alphaproteobacteria;Rhodobacterales;Rhodobacteraceae | PRJNA441682 | phycosphere | 2.15E+07 | no | no | 25% | 33% | 46.9 | 15.8 |
| Yoonia4BL | Alphaproteobacteria;Rhodobacterales;Rhodobacteraceae | PRJNA441685 | phycosphere | 6.86E+06 | yes | no | 30% | 23% | 25.6 | 5.5 |
| Pusillimonas EA3 | Gammaproteobacteria;Burkholderiales;Burkholderiaceae | PRJNA455452 | free-living | 1.31E+05 | no | no | 27% | 22% | 0.3 | 0.1 |
| Alcanivorax EA2 | Gammaproteobacteria;Pseudomonadales;Alcanivoracaceae | PRJNA455453 | free-living | 4.12E+05 | no | yes | n/a | 56% | 0.6 | 0.0 |
| Henriciella 6ES | Alphaproteobacteria;Caulobacterales;Hyphomonadaceae | PRJNA441683 | phycosphere | 1.81E+04 | b.d. | no | n/a | n/a | n/a | n/a |
| Tepidicaulis EA10 | Alphaproteobacteria;Parvibaculales;Parvibaculaceae | PRJNA455451 | free-living | 2.43E+05 | no | no | n/a | 19% | 0.1 | 0.0 |
| Roselbium PT13C1 | Alphaproteobacteria;Rhizobiales;Stappiaceae | PRJNA441686 | phycosphere | 3.79E+06 | yes | no | 33% | 24% | 3.2 | 0.3 |
| Thalassospira 13M1/11-3 | Alphaproteobacteria;Rhodospirillales;Thalassospiraceae | PRJNA441688 | phycosphere | 1.70E+06 | no | no | 30% | 36% | 1.2 | 0.1 |
| *Marinobacter 3-2 | Gammaproteobacteria;Pseudomonadales;Oleiphilaceae | PRJNA500125 | phycosphere | 2.01E+06 | yes | yes | n/a | 19% | 1.8 | 0.0 |

The summary includes enrichment source (free-living or attached), bacterial abundances at late log algal growth phase, whether co-cultures exhibited some visible attachment to algal cells, and whether algal chlorophyll in-vivo fluorescence was significantly higher at late log phase compared to axenic cultures; also included are coefficient of dispersion (CD) of the C and N incorporation (n/a indicates CD is not calculated due to no isotope incorporation detected), as well as calculations of total DOC and DON incorporated based on cell size and abundance; *cultures with two other *Marinobacter* isolates with identical 16S rRNA gene sequence also exhibited significantly higher chlorophyll fluorescence; b.d. = below detection.

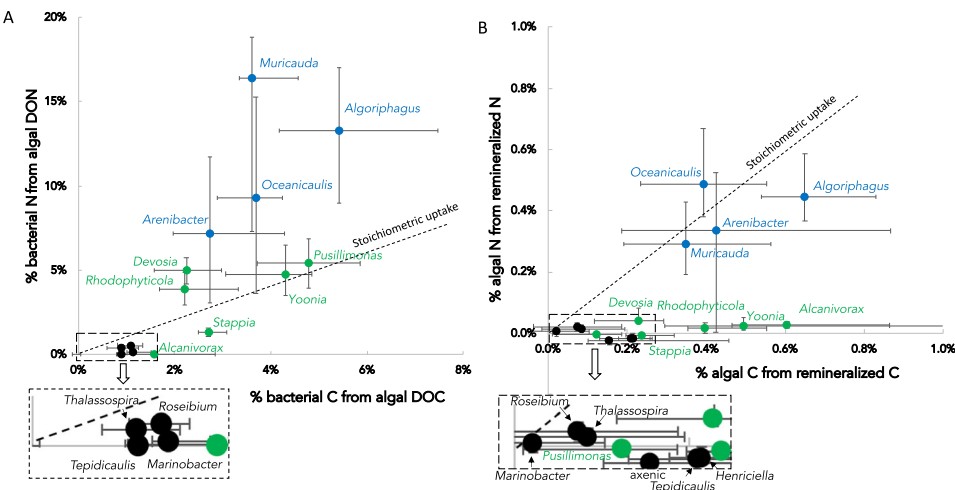

**Fig. 1 | Bacterial and algal C and N incorporation as measured by stable isotope incubations followed by nanoSIMS.** (A) relationship between C and N net incorporation for 15 *Phaeodactylum*-associated bacterial taxa (each point is the population median, error bars represent interquartile range; $R^2 = 0.55$; n = 425 total cells).

Taxa are colored blue, green, and black based on Kmeans clustering of the SIP data. (B) same relationship for algal incorporation of bacteria-remineralized C and N from the same experiments (median +/- interquartile range; n = 930 cells).

calculated the total net flux of DON into the bacterial cells using $N_{net}$, cell sizes, and abundances, finding that *Rhodophyticola* was the highest total incorporator of N with ~16 µg N L$^{-1}$ d$^{-1}$ (primarily due to its high abundances; Table 1, Supplementary data 1). Unlike DON, DOC was incorporated by all tested bacteria (compared to the killed control; Kruskal-Wallis multiple comparison test; Fig. 1A, S4). Median daily $C_{net}$ values ranged from 0.9% to 5.4%, with the highest net total volumetric incorporation from *Rhodophyticola* and *Yoonia*, the two most abundant bacteria in co-culture, at 47 and 25 µg C L$^{-1}$ d$^{-1}$, respectively. CD values for DOC incorporation ranged from 15% to 56%, with no trend between variability and incorporation. As expected, median bacterial daily $C_{net}$ and $N_{net}$ were positively correlated ($R^2 = 0.55$, p = 0.0024, Fig. 1A), also apparent at the single-cell level when including all cells (p < 0.001). Certain strains did not exhibit statistically correlated $C_{net}$ and $N_{net}$ at the single-cell level (Fig. S5), suggesting uncoupling between DOC and DON incorporation. Comparing across strains, the taxa exhibited differences in their relative incorporation of algal DON vs DOC. Six strains incorporated low DOC but little or no detectable DON (below the stoichiometric line in Fig. 1A): *Stappia, Marinobacter, Roseibium, Thalassospira, Alcanivorax,* and *Tepidicaulis*. On the other end of the spectrum, six strains met a higher percentage of their cellular N needs with algal DON relative to their C needs with algal DOC: *Muricauda, Arenibacter, Algoriphagus, Oceanicaulis, Devosia* and *Rhodophyticola*. Two strains (*Yoonia* and *Pusillimonas*) overlapped with the stoichiometric line.

Since the isotope-labeled organic matter was added to co-cultures in order to measure bacterial incorporation, this experimental design also enabled us to quantify remineralized C and N algal uptake over 24 h from the same incubations (N = 1149 total algal cells analyzed). As a population, the axenic algal cells were not significantly more isotopically enriched than the killed controls, although a portion (18%) of these cells were isotopically enriched in $^{13}$C (defined as being 3 standard deviations above the median for the killed controls; Fig. S4B). This indicates that some *P. tricornutum* cells incorporated DOC (but interestingly not DON). Thus, in order to determine isotope enrichment mediated by bacteria, we statistically compared algal cells from co-cultures to axenic cultures to test for uptake of bacteria-remineralized DOC and DON. The extent of labeling, as expected for such a relatively short incubation (24 h), was low but detectable for most of the co-cultures. For DOC, all co-cultures were significantly more enriched than the axenic (Fig. 1B, S4), but many exhibited low

levels of enrichment that we do not consider biogeochemically significant. Bacterial DOC incorporation and algal C reincorporated were positively correlated, though just above the 0.05 p-value cutoff for significance (Fig. S6A; regression analysis, $R^2 = 0.27$, p = 0.055), illustrating that greater bacterial incorporation of algal DOC was associated with greater C remineralization and subsequent incorporation by the algal cells. We note that the relatively low $R^2$ value suggests this relationship is not strongly predictive and indicates decoupling between bacterial DOC incorporation and C remineralization to algae. Regarding bacteria-remineralized N, nine of the co-cultures exhibited significantly greater algal N incorporation compared to the axenic cultures, but the majority (5) had very low incorporation which we consider biogeochemically insignificant. The other four included *Algoriphagus* (median daily $N_{net} = 0.45$%), *Arenibacter* (0.33%), *Muricauda* (0.29%), and *Oceanicaulis* (0.49%; Fig. 1B, S4B), which were of the same magnitude as the percentage of their C needs met by remineralized DOC (Fig. 1B).

One of the primary goals of this study was to use the nanoSIMS-based activity measurements in the co-cultures (both from the bacteria and the algal cells) to group bacteria into guilds with similar DOM incorporation and remineralization profiles. To accomplish this, we used a statistical K-means clustering analysis of the DOC and DON incorporation and remineralization median data, using the elbow method to identify the optimal number of clusters. This method identified three functional guilds (Fig. S6B, C; color-coded in Figs. 1, 2).

## Bacterial incorporation of low molecular weight C compounds and inorganic N

The chemical extraction protocol we used has high extraction efficiency for marine DOM but does not capture all organic compounds, especially small polar molecules such as sugars and amino acids[26]. Thus, it is likely that components of *P. tricornutum* exudates were not captured by our isotope addition experiment. To begin to address this issue, we used GC-MS exometabolite analyses to identify small polar metabolites produced by *P. tricornutum* that might be consumed by associated bacteria. We focused on *Marinobacter* as this mutualistic bacterium can lead to higher lipid accumulation in *P. tricornutum*[25] and is abundant in co-culture with many species of microalgae[27]. For these experiments, we analyzed media controls, axenic cultures, and co-cultures of *P. tricornutum* and *Marinobacter* to confirm that algal-associated bacteria can decrease the concentrations of accumulated

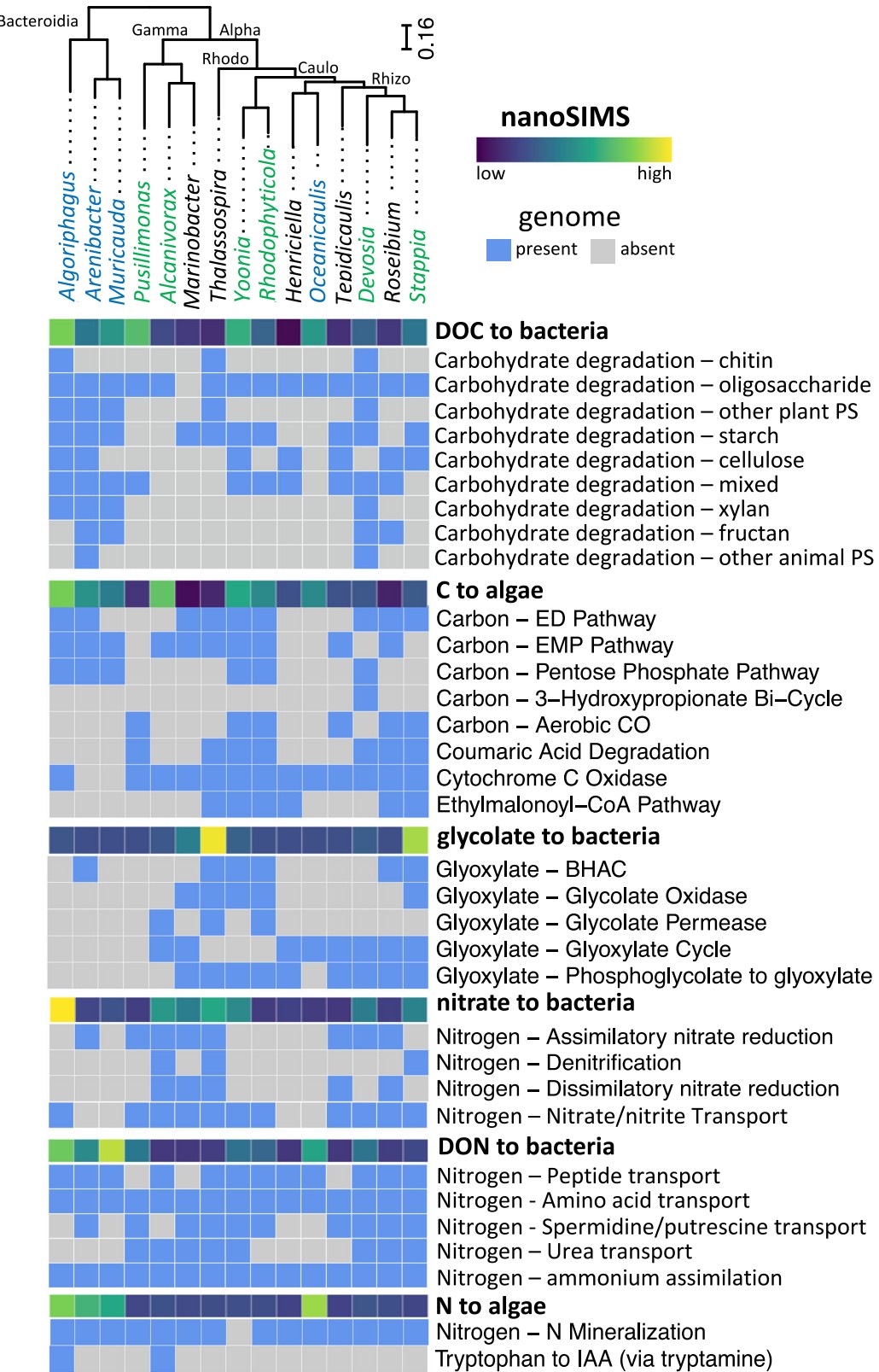

**Fig. 2 | Summary of genomic analysis of 15 *P. tricornutum* associated bacteria and comparison with nanoSIMS data (bold).** Genus names are colored according to K-means clustering from the nanoSIMS data, and bacteria are grouped based on phylogeny of single copy core genes (abbreviations: Rhizo Rhizobacterales, Caulo –

Caulbacterales, Gamma Gammaproteobacteria, Rhodo Rhodobacterales, Alpha Alphaproteobacteria; DOC Dissolved Organic Carbon, DON Dissolved Organic Nitrogen, PS Polysaccharide, ED Entner–Doudoroff pathway EMP Embden-Meyerhof-Parnas Pathway, BHAC β-hydroxyaspartate cycle).

algal-released organic matter. These analyses identified eight small metabolites in greater abundance in *P. tricornutum* cultures compared to the media controls, two of which (glucose and glycolate) were in lower abundance in *P. tricornutum-Marinobacter* co-cultures compared to axenic *P. tricornutum* cultures (Figure S7; t-test with Dunn-Sidak multiple comparison p-value adjustment). This result raises the possibility that *Marinobacter* and other strains that had low or no incorporation of larger DOC and DON molecules consumed small organic molecules (such as glucose and glycolate) excreted by *P. tricornutum* and used nitrate as an N source. Another possibility is that *P. tricornutum* released less glucose and glycolate in the presence of *Marinobacter*.

Considering these possibilities, we tested which of the 15 algal-associated strains incorporated small polar molecules with nitrate as the N source, using simultaneously added [13]C-labeled glycolate and [15]N-labeled nitrate. We chose glycolate over glucose since algal biofuel production ponds are super-saturated with oxygen during the day that lead to glycolate production via photorespiration[28]. For these experiments, we removed *P. tricornutum* cells because they incorporate nitrate and would cross-feed [15]N to the bacteria via DON release. Therefore, we added the labeled glycolate and nitrate to unlabeled *P. tricornutum* spent medium comprised of algal DOC and DON, omitting the extraction protocol. Examining glycolate incorporation, we confirmed that *Marinobacter* incorporated this algal-released compound (Fig. S8), and we qualitatively categorized the bacterial strains as high glycolate incorporators (*Stappia* and *Thalassospira*, with median 2-day $C_{net}$ > 5%), medium incorporators (*Marinobacter*, 2-day $C_{net}$ 1.5 to 3 %), low incorporators (*Alcanivorax, Algoriphagus, Devosia, Yoonia*, 2-day $C_{net}$ from 0.15 to 1%), and non-incorporators (all other strains tested; Fig. S8A). Notably, 3 of the strains (*Thalassospira, Stappia*, and *Alcanivorax*) had distinct sub-populations of glycolate non-incorporators and incorporators. The [15]N incorporation data further indicate that 9 of the strains incorporated statistically significant levels of nitrate (2-day $N_{net}$ = 0.05 to 0.9%) compared to killed controls. The average relative nitrate and glycolate use for the different strains varied by approximately an order of magnitude, with *Thalassospira, Stappia* and *Marinobacter* showing significantly more glycolate use for their C needs relative to nitrate use for their N needs (Fig. S8B). This suggests that nitrate incorporation was not enough to fully support growth based on glycolate incorporation, and the glycolate-utilizing bacteria likely incorporated unlabeled DON from the algal spent media. *Algoriphagus* incorporated the highest levels of nitrate (median $N_{net}$ = 0.87% after 2 days) but very little glycolate (median $C_{net}$ = 0.17%). Two other strains (*Muricauda* and *Arenibacter*) did not incorporate glycolate but incorporated significant but low levels of nitrate. The remaining strains (*Rhodophyticola, Pusillimonas, Roseibium, Tepidicaulis, Oceanicaulis, Henriciella*) did not incorporate nitrate, suggesting they require DON for growth. Note that *Roseibium* and *Tepidicaulis* incorporated little to no [15]N-labeled DON from the chemical extraction experiment (Fig. S4A) thus we interpret these results together to indicate that these two strains incorporated small DON molecules not captured by the chemical extraction. Due to low cellular abundances, *Henriciella* cells were not collected for analyses in the DOC-DON experiment.

## Growth without added organic matter

The isotope addition experiments suggested that some of the algal-associated bacteria, somewhat surprisingly, did not incorporate much complex algal DOM, particularly the third guild identified by the K-means clustering (colored in black on Figs. 1 and 2). These data led us to develop the hypothesis that these strains may be able to grow on background levels of organic matter not derived from algal photosynthesis. Thus, we attempted to cultivate them on agar plates without any source of organic carbon and successfully grew *Marinobacter, Roseibium, Thalassospira, Tepidicaulis*, and *Stappia* on artificial seawater agar (Fig. S9; all other strains did not form colonies on such

plates). We then tested 11 of the 15 strains in liquid artificial seawater incubated for one week in the presence of [13]C-labeled bicarbonate to quantify activity by measuring anapleurotic carbon fixation, a mechanism that heterotrophs use to replenish lost $CO_2$ in the TCA cycle. Evidence of anapleurotic activity in the absence of algal-derived organic matter would suggest metabolic activity using background levels of organic material present in the medium and not derived from algal photosynthesis. The bicarbonate uptake experiments in liquid in most cases matched the results of the growth on artificial seawater agar: strains able to form colonies on the agar media (*Roseibium, Marinobacter, Stappia, Thalassospira, Tepidicaulis*) exhibited detectable DIC incorporation in liquid with no added organic matter, and strains not able to grow on artificial seawater agar (e.g. *Algoriphagus, Arenibacter, Muricauda, Henriciella*) did not incorporate DIC (Fig. S9). *Yoonia* and *Oceanicaulis*, which did not form colonies on seawater agar, were exceptions with detectable DIC incorporation. These results show that a subset of the algal-associated bacteria exhibit detectable metabolism in the absence of algal-derived carbon.

## Genome analyses

Based on cultivation conditions, all 15 bacterial isolates are aerobic heterotrophs, but the genome analysis identified only 12 as being capable of aerobic respiration. The exceptions were the 3 Bacteroidia isolates, suggesting that genome analysis for all metabolic pathways is likely to be biased against this phylogenetic group, and any absence of other pathways within this taxon, and likely other taxa as well, should be interpreted carefully. We briefly summarize these genome analysis results as they relate to the isotope incorporation data and point the reader to the supplementary information for a longer discussion about other aspects of the genomic capabilities of these organisms, including the following metabolisms: C1-compound utilization and production, photoheterotrophy, carbohydrate degradation and synthesis, glycolate/glyoxylate utilization, plant growth hormone synthesis, degradation of algal senescence products, and vitamin production.

We first clustered the 15 bacterial strains according to phylogeny, and mapped isotope incorporation data as well as the presence of metabolic pathways relevant to algae-bacteria interactions (Fig. 2). The primary and most evident result of this analysis is that activity as measured by nanoSIMS and the resulting functional guilds do not map to phylogeny. Examining this in more detail, we did not identify links between bacterial incorporation of complex algal C and the presence of carbohydrate degradation pathways, nor between C remineralization to algae and bacterial central C metabolism pathways. We found similar lack of congruence between direct activity measurements and the presence of metabolic pathways with algal organic N incorporation, N remineralization, as well as glycolate and nitrate uptake. In most cases (but not always), there was at least one metabolic pathway (or at least a transporter) present in a genome that could explain detected activity by nanoSIMS, but activity could not be predicted by the presence of a genetic potential for specific metabolic pathways. To further examine this lack of genetic congruence with activity, we used the presence of pathways related to C and N metabolism in the genomes to form different clusters of taxa based on subsets of these pathways (Fig. S10) and again found this clustering did not match either phylogeny or clustering based on activity.

## Protein expression hints at bacterial ecological strategy

We collected proteomics data to examine the expression of bacterial proteins during co-cultivation of *Rhodophyticola* and *Marinobacter* with *P. tricornutum*. We chose to compare these two bacterial co-cultures because first, we had collected GC-MS metabolomics for *Marinobacter*, and second, they represented distinct bacterial strategies with regards to algal interactions and were grouped into different guilds by the K-means classifier: *Marinobacter* incorporates glycolate and nitrate but no complex DON, and is an algal-attached mutualist

that does not grow to high abundances, whereas *Rhodophyticola* incorporates moderate amounts of complex DON, but not glycolate or nitrate, and is a free-living commensal that grows to high abundances (Fig. S11). We identified 370 and 160 unique proteins for *Rhodophyticola* and *Marinobacter* co-cultures, respectively, based on our established cutoffs for specificity (Supplementary Data 2, Fig. S12). We examined the transporter protein annotations as they provide direct links to the nanoSIP incorporation data presented above, and the reader can find further discussion on the rest of the proteome in the supplementary information, including proteins related to oxidative stress, carbon monoxide oxidation, and denitrification. These proteomics data show the two bacteria exhibited differing strategies for organic matter assimilation, with *Marinobacter* expressing proteins for the transport of amino acids, organic and inorganic phosphorus, and sugar alcohols (hydroxyl-containing carbohydrates), and *Rhodophyticola* expressing proteins for the transport of peptides, nucleic acids, and carbohydrates (Fig. 3). Both strains expressed proteins for transport of lipids, metals, sulfur, and dicarboxylates. These data suggest that *Marinobacter* is adapted to the uptake of small organic compounds, and *Rhodophyticola* for generally larger molecules with more N content, which agree with the nanoSIP experiments presented above, keeping in mind that the chemical extraction protocol used for those experiments does not retain small molecules such as amino acids and organic acids. *Marinobacter*'s lack of $^{15}$N enrichment while expressing amino acid transporters is also consistent with the extraction protocol being suboptimal for the retention of amino acids.

## Discussion

In this study, we quantified algal DOC and DON uptake and remineralization by bacteria associated with *P. tricornutum* and used these net flux measurements to statistically categorize bacteria with similar activities into functional categories ('guilds', Fig. 4), using K-mean clustering (Fig. S5) to partition the strains. One guild, that we call macromolecule remineralizers (identified in blue on Figs. 1, 2, and 4), incorporated high amounts of complex DON and DOC and remineralized detectable levels of N and C to the algal cells. A second guild,

which we call macromolecule users (green), also incorporated both DON and DOC and remineralized detectable C, but not N to the algal cells. *Alcanivorax* is an exception in this category because it did not incorporate complex algal DON and incorporated some N from nitrate and is more similar to the third guild in that way. The third guild, which we call small-molecule users (black), incorporated complex DOC but not complex DON, likely using smaller DON molecules not captured by the extraction protocol and some N from nitrate; they also appeared to be highly efficient recyclers of lost $CO_2$ through anaplerotic fixation based on $^{13}$C bicarbonate incorporation with no algal C present. As noted above, these guilds do not map to phylogeny and are not predicted by genome content. In addition, there was no pattern regarding attachment to algal cells or mutualism. Note that these conclusions are specific for *P. tricornutum* incubated in the laboratory in high nutrient F/2 media with high cell concentrations and may not be applicable to different algal taxa or environmental conditions.

In addition to the strain-to-strain variability in activity, within each strain, we also identified cell-to-cell variability in isotope incorporation, which is commonly found even in clonal populations[29]. This variability may contribute to the ecological success of these bacteria given that they rely on organic matter from algal photosynthesis. The macromolecule remineralizers all exhibited a coefficient of dispersion (CD) for DON incorporation over 30% and the macromolecule users under 30%, suggesting that more heterogeneity in N incorporation is linked to N remineralization. One possible mechanism to explain this pattern is that remineralizers may have less efficient coupling between organic matter hydrolysis and uptake, driven by a subset of the population expressing the hydrolytic enzymes for the benefit of the entire population, including the algal cells. In this way, the algal cells and the non-enzyme-expressing bacteria are cheaters, getting benefits without expanding energy to break down the organic matter. Also, some strains included glycolate incorporators and non-incorporators within the same population. This type of heterogeneity in a clonal population has previously been hypothesized to be a response to resource limitation[30], suggesting this process could also be at play here. Another study[31], using an autotrophic clonal cyanobacterium culture, documented cell-to-cell heterogeneity caused by several other mechanisms, including temporal population differentiation in an assimilatory process, intracellular resource allocation and differential turnover rates, and heterogeneity in the uptake of multiple substrates. Here, we can consider several potential mechanisms leading to cell-to-cell

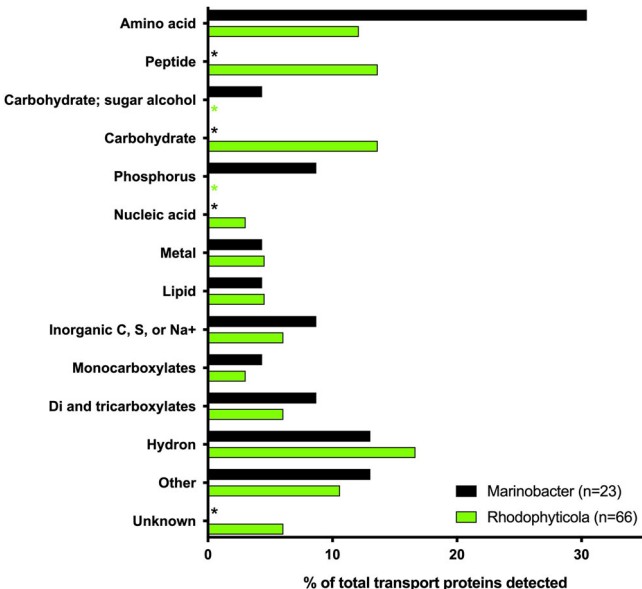

**Fig. 3 | Comparison of functional categories of annotated transporters expressed at the protein level detected from two algal-bacteria co-cultures.** Categories based on Transport DB substrate identification and classed based on ChEBI (Chemical Entities of Biological Interest) ontology. "Other" category includes proteins that had only one protein identified from that category, and include categories of: polyamine, ion, choline, detergent, osmolyte, and antioxidant. Substrate identifications and TCDB classification can be found in Supplemental Data 3.

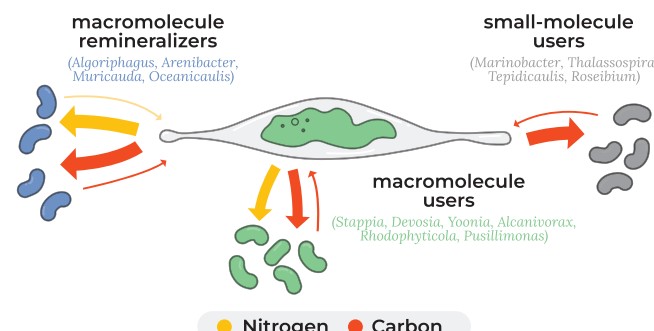

**Fig. 4 | Conceptual figure of bacterial-*Phaeodactylum* metabolic interactions, based on measurements of C and N incorporation and remineralization by 15 bacterial isolates and remineralization of C and N to** (remineralization defined as detectable transfer of remineralized algal-derived DOM to the diatom cells). Bacteria are classified as macromolecule remineralizers (evidence of nutrient feedback to the diatom cells) or users (no detected N incorporation by diatoms), and bacteria that did not incorporate complex organic N are classified as small-molecule users. Underlined genera have shown evidence of mutualism in co-culture. The thickness of the arrows approximates relative fluxes as measured by isotope incorporation.

variability in isotope incorporation by both bacteria and diatom cells. First, the cultures were in batch mode and not in chemostats and were not synchronized, so both algal and bacterial cells were in different phases of cellular division. We note that chemostat-grown cells can still exhibit cell-to-cell heterogeneity, but potentially less so than batch-mode cells[31]. Second, the cultures were not agitated, and *P. tricornutum* exists both as a benthic and pelagic organism, often forming aggregates of multiple cells, so resources might be heterogeneously distributed or differentially accessible in the culture tubes. Third, bacterial cells can be either algal-attached or free-living, which could result in differential access to externally added isotope-labeled algal organic matter, again leading to spatial heterogeneity of resources. Even among free-living bacteria, they were simultaneously incorporating the labeled extracellular algal organic matter and unlabeled organic matter being exuded from the algal cells, and the latter should not be homogeneously distributed.

Unlike DON, incorporation of DOC and remineralization of C back to algal cells was variable but statistically detectable for all tested co-cultures. Also of note is that although as a population, axenic algal cells were not statistically more enriched than killed controls, some of these cells (18%) were statistically enriched. This suggests a subset of the algal population was expressing some unusual metabolic potential to incorporate its own released DOC, through an uncharacterized mechanism. Further, this brings up the possibility that DOC-remineralizing bacteria weren't actually remineralizing C back to the algal cells but simply signaling to the algal cells to express this DOC incorporation metabolism. We do not currently have data able to fully reject this hypothesis, although the *P. tricornutum* proteome collected during co-culture with *Rhodophyticola* and *Marinobacter* was not statistically different from axenic cultures, suggesting that algal cells were in a metabolically similar state with or without bacteria.

The relationship between bacterial C incorporation and remineralized C incorporated by the algal cells in the corresponding co-culture (Fig. S6A), was positive but not strong. It likely reveals the C use efficiency of the strains on complex algal-derived organic carbon, with most taxa falling along a positive relationship: greater bacterial C incorporation corresponds to greater remineralized C. Three exceptions from this trend were *Pusillimonas* and *Marinobacter*, with low C remineralization given their C incorporation, and *Alcanivorax*, with the opposite (high C remineralization given its low C incorporation). This suggests that *Pusillimonas* and *Marinobacter* are highly efficient incorporators of complex algal C, and *Alcanivorax* is a highly inefficient one. It is likely that expanding the number of categories would split *Pusillimonas* and *Alcanivorax* into distinct groups, based on the second dimension of the K-means clustering locating them at opposite ends (y-axis in Fig. S6B). No information from previous studies with either *Alcanivorax* or *Pusillimonas* suggests they are high and low respirers of CO$_2$, respectively. However, a future study linking single-cell respiration rates[32] and C remineralization to algal cells could directly test this hypothesis.

Algal incorporation of bacterial remineralized organic matter can sometimes lead to the emergence of mutualism: for example, under long-term cultivation with low nutrient concentrations, picocyanobacterial and heterotrophic bacteria exchange organic C for remineralized nutrients[33]. Here, we defined mutualism as increased algal biomass in co-culture compared to axenic cultures at the early stationary growth phase. We did not find that the exchange of N was associated with mutualism: none of the three mutualistic strains showed evidence of N remineralization. Furthermore, the highest remineralizers of DON (*Muricauda*, *Oceanicaulis*, *Arenibacter*, and *Algoriphagus*) were not mutualistic, and two of those strains (*Muricauda* and *Algoriphagus*) were previously shown to be growth-inhibiting in co-culture with *P. tricornutum*[25]. In fact, most of the 15 tested strains, including the three mutualists, provided little to no remineralized N to the algal cells, at least from

the complex organic N extracted with the protocol used. This suggests that bacterial remineralization of complex algal DON and coupled transfer to the algal cells is not as common as we expected (Fig. S6D), at least under the relatively high nitrate concentrations tested here and commonly used in outdoor cultivation of microalgae[34]. Direct bacterial uptake of nitrate (60% of the isolates) was more prevalent than N remineralization to algal cells, although the data suggest that nitrate did not provide full bacterial N requirement for many of the taxa, based on low nitrate N$_{net}$ compared to glycolate C$_{net}$ (Fig. S7). Bacterial nitrate incorporation for biomass is more energetically expensive than ammonium, and our data confirm that the algal-associated bacteria likely did not use nitrate directly for growth. However, it is possible that algal-associated bacteria used nitrogen cycling for energy. Four strains contain the metabolic pathway for dissimilatory nitrate reduction (nitrate to ammonium), three contain genes for denitrification (nitrate to N$_2$O; Supplementary Data 3), and strains of *Marinobacter*[35] and *Thalassospira*[36] have been shown to denitrify. Furthermore, some recent evidence suggests that the same *Marinobacter* strain used here increased its growth rate in response to the availability of F/2 nutrients, likely nitrate, even in the presence of *P. tricornutum*[37]. It remains to be seen whether the mechanism is assimilatory or dissimilatory nitrate reduction, as we did not find proteomic evidence of either of these processes.

Our data and resulting conceptual framework have implications for a better understanding of C and N cycling in natural and engineered algal-dominated ecosystems. Due to the complexity of bacterial communities and the difficulty in predicting activity from 'omics data, the bacterial community is generally considered a black box with general terms for remineralization, respiration, and other biogeochemical processes[38]. Our conceptual framework should help to divide this black box into categories with shared traits, which should then help to more accurately model the impact of different microbial communities on elemental cycling. Unfortunately, our analyses suggest that activity cannot be predicted by phylogeny or the presence of metabolic pathways related to central C and N metabolism derived from genomic data. Linking genome content and activity is crucial in order to use genome data in metabolic or biogeochemical models, and future efforts should be aimed at identifying metabolic pathways that are correlated to microbial activity. We suggest that it is ultimately possible to link activity and genomic data, but that this will require identifying the unknown genomic pathways that result in these activity categories (perhaps requiring single-cell RNA or protein expression, and more comprehensive RNA/protein expression). Single-cell heterogeneity is also likely to have genomic underpinnings, and this could be another focus of future targeted research.

It remains to be seen whether the activity data resulting in the conceptual framework of bacterial metabolic interactions with *P. tricornutum* incubated in the laboratory translates to other taxa and natural communities. Nonetheless, *Phaeodactylum* is a model biofuel crop, shown to produce high amounts of biomass and lipids and can be grown in large volumes with high nutrient concentrations[39], and our results likely apply at least to this genus. Understanding the role of the different members of the *Phaeodactylum*-associated bacterial community could be useful for optimal production of algal biomass or intracellular lipids, reduction of excreted waste DOM, and recycling of nutrients after harvest. One strategy might be to optimize the algal microbiome for different phases of algal cultivation to maximize in-situ metabolic exchange during algal growth and removal of DOM from the algal-spent medium after biomass harvest. Microbiome optimization might also be a critical component of algal cultivation efforts that use recycled nutrients from wastewater or other external sources that includes both organic and inorganic nutrients.

# Methods

## Isolation and genome sequencing of bacteria

Samples originated from outdoor algal raceways at Texas A&M Agrilife that were filtered through a 0.8 micron syringe filter (to remove eukaryotes) and added to axenic *P. tricornutum* cultures[21]. These algal enrichments were further processed by removing free-living bacteria via sequential centrifugation and washing the algal cells, which should enrich for attached bacteria. Both types of samples (attachment-enriched, and original) were spread onto Marine Agar plates and colonies picked, restreaked and purified. One exception was *Henriciella* 6ES, which cannot grow on agar and was isolated via dilution to extinction. Bacterial draft genomes were sequenced from DNA extracted with a DNAEasy kit (Qiagen, Germany) at the Joint Genome Institute (JGI) through the Community Sequencing Program and annotated and initially analyzed through JGI's integrated microbial genomes (IMG) pipeline[40] and dbCAN for carbohydrate-active enzyme annotation[11]. Subsequent metabolic pathway annotation for the bacterial genomes was conducted with an approach beyond simple marker gene detection, instead assessing the completeness of pathways where multiple or alternate reactions are involved. We focused on metabolism for the cycling of substrates between algae and bacteria, specifically the pathways for either bacterial utilization of algal-derived compounds or bacterial synthesis of compounds that influence algal physiology (Supplementary data 3). The database included 359 genes from 55 distinct pathways for biogeochemical reactions related to carbon and nitrogen metabolism, other reactions of primary metabolism (photoheterotrophy, purine, and pyrimidine synthesis), secondary metabolism (cell-cell communication and signaling, degradation of aromatic amino acids, vitamin synthesis), and breakdown of polysaccharides and protein. As some genomes were draft quality and not fully complete, we used a threshold of 75 % of the pathway or greater being present to consider a pathway as functionally complete. To compare genome information to activity measurements, we clustered genomes (hclust with complete linkages in R) according to the presence of the metabolic pathways mentioned above. Code available at https://github.com/jeffkimbrel/gator.

## Cultivation conditions and growth quantification

All experiments were carried out on axenic and co-cultures generated as follows. Bacterial strains were each reinoculated individually with a sterile loop from a single colony (except *Henriciella* which does not form colonies and was isolated via dilution to extinction) into previously axenic cultures of *P. tricornutum* CCMP 2561 (National Center for Marine Algae and Microbiota; ncma.bigelow.org) and these cultures were grown in batch mode and serially transferred into new media monthly for at least 6 transfers before the start of all experiments. This approach was intended to generate long-term algal-bacterial co-cultures in balance, with bacterial population abundances dependent on algal-released organic matter rather than that from the high nutrient media used for their isolation. Axenic *P. tricornutum* cultures were determined to be bacteria-free via fluorescence microscopy after DAPI staining, as well as sequencing of the 16S rRNA gene[24], and bacterial attachment was enumerated via fluorescence microscopy from 20-day old, early stationary phase co-cultures, which has been shown to be the growth phase with maximum bacterial attachment[41]. Free-living bacterial abundances were quantified for bacterial cell abundance on an Attune benchtop flow cytometer with a CytKick autosampler (Thermo Fisher Scientific, Waltham, MA) fitted with a blue laser (excitation 488 nm). One mL of each sample was collected and immediately fixed with glutaraldehyde (0.25% final concentration), flash frozen, and stored at −80 °C. Fixed samples were diluted in sterile-filtered media to achieve >4000 event counts per second and stained with 1X SYBR Gold (Thermo Fisher Scientific, Waltham, MA) for 10 minutes in the dark prior to analysis. The instrument parameters were set up to threshold at 0.1 x 1000 on the BL1 detector, sample acquisition volume of 200 μL run at a flow rate of 100 μL/min. Gating was performed based on forward scatter, side scatter, and fluorescence of SYBR Gold (BL1 detector, emission/BP of 530/30). MilliQ blanks were run between treatment replicate sets to reduce carryover, and media blanks were included to correct for background particle noise.

Co-cultures and the axenic culture were maintained in borosilicate glass 13 mm diameter tubes on a 14/10 hr light/dark regime at 75 μmol quanta $m^{-2} s^{-1}$ (cool white fluorescent) at 22 °C. The seawater (F/2) medium was prepared using Instant Ocean salts (35 g/L) with 880 μM nitrate[42]. To test if the bacterial strains could impact algal biomass accumulation and relate it to nutrient remineralization measured by nanoSIMS in experiments described below, we tested the influence of bacterial co-cultivation on algal biomass in 96 well plates (in triplicate) under temperature and nutrient conditions at which they were maintained (22 °C, F/2). As a proxy for biomass, we compared invivo chlorophyll fluorescence (excitation 440 nm, emission 680 nm) collected on day 18 to axenic cultures with a one-way ANOVA, adjusting for multiple tests with the Dunn-Sidak method.

## Isotope tracing experiments and nanoSIP analyses

To quantify bacterial assimilation and remineralization of algal organic matter, we designed a set of experiments where algal-bacterial co-cultures were incubated in the presence of $^{13}C$- and $^{15}N$-labeled algal exudates, obtained through solid phase extraction (SPE)[43] of axenic *P. tricornutum* spent medium. We grew axenic *P. tricornutum* in 99 atom % $^{15}NO_3^-$ (Cambridge Isotopes) and ~50 atom % $^{13}CO_2$, added as 2 mM sodium bicarbonate (99% $^{13}C$, Sigma Isotec) into background 2.1 mM DIC (unlabeled) seawater media, in a sealed glass bottle to avoid further dilution of $^{13}C$ from atmospheric $CO_2$. The culture was grown until the stationary phase and filtered through a 0.2 μm bottle-top filter (Corning) and we used SPE of the labeled filtrate with columns previously shown to extract > 50 % of natural marine organic matter[43]. Briefly, the filtered exudate was acidified with hydrochloric acid to pH = 2, run through a Bond Elut PPL column (Agilent), rinsed, and eluted in methanol. This labeled material (added at a 2X final concentration to make up for 50 % loss of the extraction protocol) was added to in-balance *P. tricornutum* co-cultures (and the axenic culture, described above) in biological triplicates and incubated for 24 h under a 14/10 hr light/dark cycle. No-addition and formalin-killed co-culture controls were also included, and for subsequent calculations of substrate incorporation[44], we quantified the $^{13}C$ and $^{15}N$ enrichment of the added exudate (measured as 18 % $^{13}C$ and 45 % $^{15}N$) by nanoSIMS after spotting on a Si wafer. For a second isotope tracing experiment to examine the direct incorporation of nitrate in the presence of a carbon source, we tested the bacterial isolates without *P. tricornutum* cells incubated in unlabeled *P. tricornutum* spent medium. We added 500 μM glycolate (99 % $^{13}C$, Sigma Isotec) and 800 μM nitrate (98 % $^{15}N$, Sigma Isotec) for 48 h to triplicate cultures, which were in-balance *P. tricornutum*-bacteria co-cultures filtered through a 0.8 μm syringe filter (so algal cells were removed, but spent medium were retained). For a third isotope tracing experiment, we tested the anapleurotic activity under extremely low nutrient conditions by incubating the strains in autotrophic medium (artificial seawater ESAW) with no added carbon plus $^{13}C$-labeled bicarbonate (1.5 mM added) for one week. Results of the third experiment are only available for 11 of the 15 strains because 4 strains showed signs of contamination. Cultures were fixed with 2 % formaldehyde and filtered onto 0.2 μm white polycarbonate filters (Whatman Nucleopore, GE Healthcare Life Sciences, Pittsburgh, PA). Filters were washed with 0.2 μm filtered water, air dried and wedges of 1/8 size were cut out of each filter using sterile scissors and adhered to aluminum disks using conductive tabs (#16084-6, Ted Pella, Redding, CA) and sputter coated with ~5 nm of gold. Isotope imaging was performed with a CAMECA NanoSIMS 50 at Lawrence Livermore National Laboratory. The primary $^{133}Cs^+$ ion beam was set to 2 pA,

corresponding to an approximately 150 nm diameter beam diameter at 16 keV. Rastering was performed over 20 x 20 μm analysis areas with a dwell time of 1 ms pixel$^{-1}$ for 19–30 scans (cycles) and generated images containing 256 x 256 pixels. Sputtering equilibrium at each analysis area was achieved with an initial beam current of 90 pA to a depth of ~60 nm. After tuning the secondary ion mass spectrometer for mass resolving power of ~7000 (1.5x corrected), secondary electron images and quantitative secondary ion images were simultaneously collected for $^{12}C_2^-$, $^{13}C^{12}C^-$, $^{12}C^{14}N^-$, and $^{12}C^{15}N^-$ on individual electron multipliers in pulse counting mode; note that $^{13}C^{12}C^-/^{12}C_2^- = 2 \cdot {}^{13}C/^{12}C$ and $^{12}C^{15}N^-/^{12}C^{14}N^- = {}^{15}N/^{14}N$. All NanoSIMS datasets were initially processed using L'Image (http://limagesoftware.net) to perform dead time and image shift correction of ion image data before creating $^{13}C/^{12}C$ and $^{15}N/^{14}N$ ratio images, which reflected the level of $^{13}C$ and $^{15}N$ incorporation into biomass. To quantify substrate incorporation, cells were identified based on $^{12}C^{14}N^-$ images, and regions of interest (ROIs) were drawn manually around each algal cell, making sure to avoid including regions of attached bacterial cells, which in some cases were more highly enriched than the algal cells. ROIs for bacterial cells were drawn separately with the automated algorithm using the $^{15}N$ isotope ratio image. For co-cultures that did not incorporate algal organic matter and thus could not be identified by the $^{15}N$ isotope ratio image, ROIs were drawn manually around free-living (i.e. not associated with algal cells) bacteria-size particles using the $^{12}C^{14}N^-$ image. The isotopic ratio averages and standard error of the mean for each ROI were calculated from ratios by cycle. The ROI data were exported for further statistical analyses in GraphPad Prism.

In order to calculate how much C or N a cell incorporated from a substrate that was isotopically labeled, we calculated $X_{net}$, either $C_{net}$ and $N_{net}$[19,44]. These represent the fraction of a cell's biomass that originated from the added isotope labeled substrate, and takes into account the isotope labeling at the beginning of the incubation (formalin killed controls), the level of isotope labeling of the substrate (either obtained from the manufacturer or directly measured), and any unlabeled natural abundance background substrate. For the algal organic matter experiment (mentioned above), we directly measured the isotope composition of the extracted labeled organic matter. Based on the literature, the concentration of DOM during log phase is about half of a stationary phase *P. tricornutum* culture, so we calculated that the isotope-labeled added DOM would be further diluted by 50%. We also note that since the same amounts of algal organic matter was added to all the co-cultures, any variation from this calculation would similarly affect the coculture data and would not impact our conclusions. To calculate $X_{net}$, cellular isotopic ratios measured by nanoSIMS were converted to the fraction of the less abundant isotope in the measured biomass e.g., for $^{15}N$, $f_x = [{}^{15}N/({}^{15}N + {}^{14}N)]$. $X_{net}$ % is a measure of the newly synthesized biomass relative to the final biomass for a given element X[45], and uses the initial measurement of the cells prior to incubation (or killed control; $f_{X_i}$), the measurement at the final timepoint ($f_{X_f}$), and the isotope fraction of the substrate ($f_{X_s}$):

$$X_{net\%} = \frac{f_{X_f} - f_{X_i}}{f_{X_s} - f_{X_i}} \times 100\% \qquad (5)$$

To compare across strains, we calculated $C_{net}$ and $N_{net}$ % per day, using the formalin killed controls to represent the initial enrichment of the cells, and the isotope labeling of the added substrates, including any dilution from unlabeled substrate. For the dual-labeled algal organic matter additions (directly measured) that were added in the same concentrations to all co-cultures, our calculations took into account a further dilution by the presence of 50% unlabeled organic matter released by the algal cells. For the nitrate and glycolate experiment, we assumed that unlabeled glycolate and nitrate in the spent medium from the early stationary phase was minimal compared to what was added and thus the labeled substrates were undiluted.

These assumptions were based on previous publications documenting total dissolved organic carbon in a spent medium of 300 μM[46] and high drawdown of nitrate[47] in *P. tricornutum* cultures. Single-cell data from biological triplicate incubations were pooled to compare co-cultures versus axenic cultures or killed controls with a non-parametric Kruskal-Wallis test, followed by a Dunn's test with multiple comparison p-value adjustment via the Dunn-Sidak method. NanoSIP median data from the DOM incubations were used to cluster bacteria into groups using K-means with an optimal cluster count of three as determined by the elbow method in the R factoextra package (note that due to missing data, *Henriciella* was removed from this analysis). To quantify cell-to-cell variability within a culture, we calculated the quartile coefficient of dispersion using the first ($Q_1$) and third ($Q_3$) quartiles, defined as ($Q_3$-$Q_1$)/($Q_3 + Q_1$). Total algal DOC and DON daily incorporation into each bacterial strain was calculated using bacterial abundances in the co-cultures, the median Cnet and Nnet daily % from the populations, and cellular C and N content estimated from cell size[48] from the nanoSIMS data. These calculations are likely to be underestimated due to our assumption that biomass accumulation of single cells was linear rather than exponential during the length of the incubations[49].

## GC-MS metabolomics

The SPE protocol used to extract isotope labeled *P. tricornutum* exudate is known to have low extraction efficiency for small polar metabolites[26], and thus our isotope tracing experiments overlooked exchange mediated by these molecules. To examine bacterial incorporation of such small metabolites, we carried out GC-MS analyses of spent media from axenic *P. tricornutum*, axenic *Marinobacter* strain 3-2, media blanks, and *P. tricornutum-Marinobacter* co-cultures (all incubated in F/2 media). Triplicates from all 4 treatments were grown in 50 % seawater salinity (16 g Instant Ocean L$^{-1}$), which increased the detection of metabolites by decreasing salt contamination, collected in the early stationary phase (after 10 days of incubation) by filtration through 0.2 μm membrane filters (Corning) and frozen at −20 °C. Spent media samples containing extracellular metabolites were dried down completely in a speed vacuum concentrator (CentriVap, LAB-CONCO, Kansas City, MO). The dried metabolites were chemically derivatized by two-step derivatizations as reported previously[50]. The derivatized metabolites were analyzed by gas chromatography-mass spectrometry (GC-MS) system (Agilent Technologies, Inc. Santa Clara, CA). All the collected mass spectrometer data files were converted to netCDF format and processed using Metabolite Detector[51]. All the peaks were matched to PNNL's augmented version of the Agilent Fiehn metabolomics database, which has a retention index and fragmented spectra of metabolites, and additionally cross-checked with NIST20 and Wiley 11th Edition GC-MS databases. Identification of detected metabolites was validated manually to avoid misidentification of metabolites or false positive and negative errors. Peak area values of identified metabolites were compared among treatments.

## Proteomics

We also examined the expression of *Rhodophyticola* 6CLA (relatively high cell abundances of >10$^7$ cells mL$^{-1}$) and *Marinobacter* 3-2 (relatively low abundances of ~10$^6$ cells mL$^{-1}$) bacterial proteins in the presence of *P. tricornutum*. Three *P. tricornutum* cultures including (1) axenic, (2) co-culture with *Marinobacter* 3-2, and (3) co-culture with *Rhodophyticola* 6CLA were acclimated and maintained for at least 3 rounds of growth in fully defined artificial seawater medium[52] with modified nutrient levels (1.4 mM NO$_3$ and 58.7 μM PO$_4$). All cultures were maintained at 22 °C, 75 μmol quanta m$^{-2}$ s$^{-1}$, and a 14/10 light/dark cycle. Cultures were then scaled up to 200 mL volume, grown to late exponential phase, and used to inoculate five replicate 500 mL cultures, which were incubated for 7 days (cultures were harvested at day 5). Daily samples were collected for growth and microscopy – fluorescence was measured on 1 mL subsamples, another 1 mL fixed with 4%

paraformaldehyde, and reserved for bacterial microscopy counts. After 5 days, all cultures were harvested, with all flasks processed in parallel to avoid growth phase differences in expression, reserving 10 mL in the flasks returned to the incubator for a final fluorescence measurement on day 7 to confirm mid-exponential growth phase at time of harvest and track bacterial growth via microscopy on fixed samples. The sampled volume (approximately 485 mL) was centrifuged at 3000 x $g$ for 5 min at 4 °C, pellet resuspended in 4 mL sterile ESAW, aliquoted into two 2.0 mL microcentrifuge tubes, and centrifuged for 5 min at 3000 x $g$ at 4 °C. This second supernatant was decanted and pellets snap frozen in liquid nitrogen and transferred to −80 °C for storage.

Bacterial microscopy counts were conducted by diluting fixed samples 10-fold with 0.2 μm filtered water and filtering on 0.2 μm GTBP filters (MilliporeSigma, Burlington MA), mounting on glass slides with 13 μl slide mountant with 5 μg/μl DAPI[53] and acquiring 10 images (max intensity z-stack per filter on an inverted fluorescence scope (Dmi6000B, Leica Microsystems, Buffalo Grove IL) with a 100X oil objective, bacteria manually counted on each image, and converted to bacterial cells/mL by multiplying the mean number of cells per field of view by the total filter area divided by the field of view area divided by the volume filtered[54].

Samples were transferred to 2 mL snap-cap centrifuge tubes (Eppendorf, Hamburg, Germany) with 0.1 mm zirconia beads and bead beat in a Bullet Blender (Next Advance, Averill Park, NY) at speed 8 for 3 minutes at 4 °C. After bead beating the lysate was spun into a 15 mL Falcon tube at 2000 x $g$ for 10 minutes at 4 °C. Each sample was transferred to new tubes and a bicinchoninic acid (BCA) assay (Thermo Scientific, Waltham, MA USA) was performed to determine protein concentration. Urea was added to each tube to bring the concentration to 8 M (Sigma-Aldrich, Saint Louis, MO) and dithiothreitol (DTT) was added to the samples at 10 mM. The samples were incubated at 60 °C for 30 minutes with constant shaking at 800 rpm. Samples were then diluted 8-fold for preparation for digestion with 100 mM $NH_4HCO_3$, 1 mM $CaCl_2$ and sequencing-grade modified porcine trypsin (Promega, Madison, WI) was added to all protein samples at a 1:50 (w/w) trypsin-to-protein ratio for 3 h at 37 °C. Digested samples were desalted using a 4-probe positive pressure Gilson GX-274 ASPEC™ system (Gilson Inc., Middleton, WI) with Discovery C18 100 mg/1 mL solid phase extraction tubes (Supelco, St. Louis, MO), using the following protocol: 3 mL of methanol was added for conditioning followed by 2 mL of 0.1 % TFA in $H_2O$. The samples were then loaded onto each column followed by 4 mL of 95:5: $H_2O$:I, 0.1 % TFA. Samples were eluted with 1 mL 80:20 I:$H_2O$, 0.1 % TFA. The samples were concentrated down to ~100 μL using a Speed Vac and a final BCA was performed to determine the peptide concentration and samples were diluted to 0.1 ug/uL with nanopure water for MS analysis.

A Waters nano-Acquity dual pumping UPLC system (Milford, MA) was configured for on-line trapping of a 5 μL injection at 5 μL/min with reverse-flow elution onto the analytical column at 300 nL/min. Columns were packed in-house using 360 μm o.d. fused silica (Polymicro Technologies Inc., Phoenix, AZ) with 2-mm sol-gel frits for media retention and contained Jupiter C18 media (Phenomenex, Torrence, CA) in 5 μm particle size for the trapping column (150 μm i.d. x 4 cm long) and 3 μm particle size for the analytical column (75 μm i.d. x 70 cm long). Mobile phases consisted of (A) 0.1 % formic acid in water and (B) 0.1 % formic acid in acetonitrile with the following gradient profile (min, %B): 0, 1; 2, 8; 20, 12; 75, 30; 97, 45; 100, 95; 110, 95; 115, 1; 150, 1.

MS analysis was performed using a Q-Exactive HF mass spectrometer (Thermo Scientific, San Jose, CA) outfitted with a home-made nano-electrospray ionization interface. Electrospray emitters were prepared using 150 um o.d. x 20 um i.d. chemically etched fused silica[55]. The ion transfer tube temperature and spray voltage were 320 °C and 2.2 kV, respectively. Data were collected for 120 min following a 20 min delay from sample injection. FT-MS spectra were acquired from 350–2000 m/z at a resolution of 30 k (AGC target 1e6) and while the top 12 FT-HCD-MS/MS spectra were acquired in data dependent mode with an isolation window of 2.0 m/z and at a resolution of 15 k (AGC target 1e5) using a normalized collision energy of 30 and a 45 sec exclusion time. For each of the co-cultures there were five biological replicates and two technical replicates for a total of 10 runs per co-culture. Bacterial proteins were not abundant relative to the algal proteins, and there were a few peptides that were detected in either the axenic control culture or the incorrect co-culture (e.g. a *Marinobacter*-predicted peptide in the *Rhodophyticola* co-culture). To account for the low abundance and the non-specificity, we limited our analysis to proteins detected in both technical replicates of at least one of the five biological replicates, and proteins that at least 80% of protein counts were detected in the correct samples, which typically allowed for one peptide detected in one of the 20 non-specific sample runs.

## Reporting summary
Further information on research design is available in the Nature Portfolio Reporting Summary linked to this article.

## Data availability
Bacterial strains are available from the lead author upon request. The microbial genome data generated in this study have been deposited in NCBI's sequence read archive (SRA) and JGI's Integrated Microbial Genome (IMG) database under accession codes available in Supplementary Data 1. Genome analysis code available at https://github.com/jeffkimbrel/gator. The proteomics data have been deposited in PRIDE under accession PXD036252. All data generated in this study are provided in the Supplementary Information/Source Data file. Source data are provided with this paper.

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

## Acknowledgements

Thanks to Arnechia Harper (Georgetown University summer intern) for bacterial microscopy counts. Part of this research was carried out at Lawrence Livermore National Laboratory (LLNL) under Contract DE-AC52-07NA27344. X.M., T.J.S., J. A.K., M.M.M., K.R., C.S., C.R., R.K.S., and P.K.W., as part of the LLNL Biofuels Science Focus Area FWP SCW1039 (received by R.K.S.), were supported by the Genome Sciences Program of the U.S. Department of Energy's Office of Biological and Environmental Research. A portion of this research (supporting Y.-M.K., N.M.-M., C.N., S.P., and M.L.) was performed on project award 50220 (received by T.J.S.) under the FICUS program and the Joint Genome Institute (JGI) CSP (award 1939, received by X.M.) and used resources at the JGI and the Environmental Molecular Sciences Laboratory (EMSL), which are DOE Office of Science User Facilities. Both facilities are sponsored by the Biological and Environmental Research program and operated under Contract Nos. DE-AC02-05CH11231 (JGI) and DE-AC05-76RL01830 (EMSL).

## Author contributions

X.M. initiated the study and directed the project. X.M., T.J.S., K.R., C.S., and C.R. performed experiments and prepared samples for analyses, J.A.K. and M.M.M. performed bioinformatics analyses, M.L., C.N., and S.P. performed proteomics, Y.-M. K. and N. M.-M. performed metabolomics, X.M., P.K.W., and R.K.S. interpreted data, X.M. wrote the initial draft, and all authors revised and approved the manuscript.

## Competing interests

The authors declare no competing interests.
