## [Peer Review File · Nature Communications]

Single cell isotope tracing reveals functional guilds of bacteria associated with the diatom *Phaeodactylum tricornutum*Reviewer #1 (Remarks to the Author):

The authors present an intricate and extensive dataset obtained via a series of isotope-labeling incubation experiments involving one diatom species (*Phaeodactylum tricornutum*) and 15 diatom-associated bacterial species. The results include data on C and N assimilation obtained with a single-cell resolution (by nanoSIMS) as well as genomic, proteomic, fluorescence, GC-MS, and cell count data obtained on bulk samples. Based on this dataset, they propose functional categorization of the diatom-associated bacteria into "macromolecule remineralizers", "macromolecule users", and "small-molecule users". They also suggest that this categorization can, ultimately, be useful in mechanistic models (conceptual but also numerical) of C and N cycles in aquatic environments.

Overall, I find both the experimental design and data interpretation to be sound and well executed. Especially the number of analyses, and their diversity, are very impressive. Also, the final conceptual synthesis of the results, as proposed by the authors, is useful outside the scope of this study. However, as outlined below, I feel there are quite a few aspects that need to be clarified before the manuscript can be considered for publication.

1. The categorization is based, essentially, on the single-cell ¹³C and ¹⁵N data obtained by nanoSIMS. The power of this approach is that it reveals the pronounced cell-to-cell heterogeneity within populations that are typically considered as "homogeneous". This is well demonstrated by the present data, which shows that although a group of cells is thought to be the same in one respect (e.g., because they belong to a specific bacterial species or have a specific genomic make-up), the activity (quantified through ¹³C and ¹⁵N assimilation) substantially differs among the individuals. This raises a larger question: how useful is the categorization into groups, if individuals (and likely also many individuals rather than just a few) from one (functional) category behave, or appear to behave during a given experiment, as members of another (functional) category?

I am bringing up this issue because I feel that the authors should touch upon intercellular heterogeneity revealed in the nanoSIMS data, but they don't. There are several issues to address.

1a. Why is there such a large intercellular heterogeneity within one bacterial species (Fig. S4A & S7A) in the first place? The same question applies to the diatom species (Fig. S4B).

1b. Was the clustering based on Kmeans derived from ¹³C and ¹⁵N data of *all* individual cells, or only using the median values derived for each species? Figure S5-B suggests the latter, but I wonder whether this is a correct approach. It would be useful to see how the different individual bacterial cells fall (or not) within the depicted cluster boundaries. It also raises a bigger question related to the one above: how well defined, and thus meaningful, *are* the boundaries?

1c. Is the median value the right parameter to use when comparing the nanoSIMS data with other data? For example, genomic and proteomic data are done on bulk cell biomass, so represent an "average cell". But the median may not be a good representation of the average cell if the distribution of the ¹³C or ¹⁵N enrichment is skewed (which it seems to be for most species) or shows a clear division into two (or perhaps more) sub-populations (Fig. S4A, S7A).

1d. This inconsistency between the mean and median values implies that the comparison of genomic/proteomic data with nanoSIMS data may not be as meaningful as we would like it to be, especially if we do not understand the reason behind the large intercellular heterogeneity seen in the nanoSIMS data and because we do not have the corresponding genomic/proteomic data for each individual cell that was measured by nanoSIMS (this is a more general problem, not specific to this study). To put it more bluntly, it is unclear what we are categorizing here, and why it is meaningful: individual cells, or individual cells that have already been "pre-categorized" (e.g., by assigning them to species, or by calculating a median for a population)?

2. The authors chose to search for links between phylogeny (as derived from 16S rRNA) and activity (or a type of ecological interaction derived from the activity), but found none. However, the same conclusion can be made for the larger genomic (Fig. 2, squared diagrams at the bottom), or even proteomic (Fig. 3), dataset when compared with the nanoSIMS dataset. We simply cannot

predict from the genomic potential, or even the proteomic make-up, of an *average* cell how active an *individual* cell will be in terms of C and N assimilation. I think this gap should be more emphasized by the authors (they mention it gently in the abstract, l. 31-32, but I think it deserved a bigger discussion).

3. Given the exhaustive analysis of cells by nanoSIMS and genomics (15 species), the proteomic analysis on 2 bacterial species feels somewhat unsatisfying. I understand it's a lot of extra work, but tremendous amount of work has been put into this study already. The question is, however, whether we could actually learn something from the proteomics data in the context of the aim of this study. For example, would the proteomic data be sufficiently strong to allow prediction of the ecological interaction between the diatom and a given bacterial species (i.e., assignment to one of the three categories defined by the authors)? Looking at the proteomic profiles for the 2 species shown in Fig. 3, I am not sure. Actually, the usefulness of the proteomic is not quite clear to me given the fact that 3 functional categories have been identified by the authors, whereas representatives from only 2 categories were studied in greater detail.

===

I provide the rest of my comments as they come to my mind, which roughly follows the main text (line numbers indicated), because ordering based on priorities (more vs. less important) would be too exhausting.

1-2: The title makes too broad a statement while the results are limited to one diatom species, and even under specific culturing conditions (e.g., in an F/2 medium). Given point 2 above, I don't find it useful to emphasize the uncoupling to phylogeny in the title. In fact, the authors should clarify why the existence of a link between phylogeny and the magnitude of the diatom-bacteria exchange of C and N would be a viable hypothesis (to me, it is not). It would be better to make the title more neutral and to the point of the study, something like "Single cell carbon and nitrogen incorporation and remineralization profiles of bacteria associated with the diatom *Phaeodactylum tricornutum*", or perhaps something shorter.

32-35: Yes, but the data is insufficient to generalize this statement (only 2 species compared). Also, looking at the overall proteomic profiles, it seems difficult to see that they could be used to predict the categorization of the bacterial strains.

35-37: I can fully identify myself with this statement. But how will the categorization be done to make it practical in situations mentioned in the subsequent sentence (i.e., for models)? For now, only genomic (and perhaps proteomic) based categorization of (natural=mixed) microbial communities can be done with a sufficiently high throughput and sensitivity/resolution. The authors basically show that those type of results are more or less "useless" if we are to understand the ecological interactions within the community in terms of C and N fluxes - because they cannot predict categorization even into the three major functional categories identified by the authors. In other words, the proposed categorization may be useful for modellers, yes (l.38-39), but how will they constrain them based on practically available data?

50-53, 57, 72-73: C and N fluxes is not quite what the authors are presenting here; they present the relative amount of assimilated C and N, as % of the cellular C and N content. Thus, these two formulations prime the reader with somewhat "false" expectations about what is done in this study. I suggest to reword these parts to better reflect the content and focus of the present study.

54: as written, the sentence does not make sense.

95: Not only identity, but also the larger genetic potential of the studied bacterial species.

99: more accurately, not what is "made available", but what is actually assimilated by (transferred to) the algal cell.

101: be more clear about "representative". In which sense?

103-105: The authors formulate these hypotheses as a main goal of the study, but do not make firm statements about their validity. The correlation hypothesis for C transfer is confirmed, but presented in a rather hidden way (as Fig. S5-C and one sentence). No obvious statement is made about the correlation hypothesis for N transfer.

107: not "metabolic function" but "metabolic potential".

110-111: I am not sure how to read this sentence (especially the part after the comma, including the comma).

113: I am not quite sure how the labeling of the C and N sources was derived/measured from this experimental design (addition of isotope-labeled P. tricornutum-derived organic matter to these co-cultures). How was the isotope-labeled P. tricornutum-derived organic matter prepared is also not described.

119: "These" data... Which data? All together, or only those mentioned in the previous sentence?

121: I think the use of "showing" here overstretches the applicability of the results. Especially because the study only focuses on 1 algal species under specific "environmental" conditions (e.g., high nitrate availability). I would agree that the three categories are a useful suggestion for conceptual models, a better first approximation of algal-bacteria interactions than the one currently commonly used.

128-133: Very long sentence, difficult to follow. Consider splitting to make it easier to follow.

143: 75% - percentage of what exactly?

150: unclear, too brief. transferred where?

155: add hr to the light/dark cycle.

156: Please be explicit about the nitrate concentration here, as it is not immediately obvious - at least to a reader without practical knowledge of cell culturing - that F/2 medium is rich in nitrate. It will be important later on when you mention that the conclusions in this study are relevant when nitrate is available as a source of N.

157-159: Is this describing what is shown in Sup Fig S2? If so, it is unclear how the ANOVA was done, and which "multiple tests" it was "adjusted for". Specifically, what exactly is compared in Fig S2? The last time points, or all time points? When I look at it, the curves all look rather similar and of two rough "types" (similar to either Marinobacter or Devosia). Thus, it is not clear why the curves shown in the lower-right panel show no significant influence of the bacteria on the diatom growth, while the other panels do show significant influence.

163: Add "assimilation and" before "remineralization".

165: Add SPE after "solid phase extraction", otherwise the abbreviation is unclear later in the text. Also, clarify how the labeled biomass was obtained in the first place (with a little more detail than just with a citation).

168: add "hr" for the light/dark cycle.

171: You determined the labelling of the added exudates, but what was the labelling of the exudates in the actual incubation? Why wasn't that measured instead? If it were, the assumption about the dilution of the added exudates by the exudates present in the co-cultures would be unnecessary. The information about the substrate labeling is required when calculating the Cnet and Nnet later on. The same for the second experiment (l. 175-177).

178: Why only 11 and not all 15? What determined the choice?

181-184: Calculations not clearly described. I am familiar with your previous work, so I can guess what you did here, but for an average reader this description is insufficient. A reference to a previous calculation approach should at least be added, but preferably a more detailed explanation. Of particular importance is to be clear about the labeling of the substrates.

At this point, I also wonder why you only convert the enrichment data into the relative amounts of assimilated C and N (Cnet and Nnet in %) and not to relative rates (%C or %N per day) or even cell-specific rates (mol C cell⁻¹ day⁻¹ or mol N cell⁻¹ day⁻¹). Using the rates would be more appropriate considering that C and N *fluxes* (which are expressed per unit time) are your stated aim of the study. Additionally, relative rates are useful from the perspective of the bacteria (they reflect cell's growth rate on a specific substrate, which is added as a labelled tracer during the SIP experiment), but a cell-specific rate is the relevant quantity when describing an ecological interaction (C or N flux) between algae and bacteria in models. Since this is the context and aim of your study, it may be even necessary that you determine the cellular C and N contents of the bacteria (and diatom) and convert your nanoSIMS data into cell-specific rates. In this way, the differences between the studied bacterial species may decrease or become more pronounced, depending on the differences in their cellular C and N contents and C:N ratios. I wonder whether the categorization of bacteria would differ if cell-specific rates were used instead of the relative amounts of assimilated C and N (Cnet and Nnet). Considering these comments, please include a short discussion about your preferred choice of nanoSIMS-derived quantities as a measure of ecological interactions between bacteria and algae.

185-186: Please explain more clearly. How does 66% labeled 33% unlabeled "turn" into 50%? It is not clear to me what you are talking about here.

190: calculated -> assumed

192: Unclear. 300 mM is >> 500 uM glocolate added.

193: "to compare across treatment". To compare what?

195-197: Was the K-means clustering done on nanoSIMS data of individual cells, or using the median (or mean) value for each species? If the latter, how was the cell-to-cell heterogeneity, which differed greatly among the different species, accounted for?

206: Unclear why only these 2 species were selected for this analysis.

220-223: Unclear how this statement is supported by data shown in Fig. S2. I don't know where to look. As mentioned earlier, all curves look very similar to me and of two types.

230: replace to "could not be"

231: It would be useful to add here also the algal cell counts. Note that Table 1 does not specify the units of the cell counts.

242: Yes, high variability among isolates. However, there is also high variability among individual cells of a given isolate, and this is not touched upon at all. Can we understand variability among isolates if we do not fully understand variability within isolates?

249: not Devosia?

253: I assume you are talking here about a correlation at the level of median values for each species. But was there also correlation on the level of individual cells? That is, were Cnet and Nnet in cells correlated for a given species, and consistently so for all species?

259: What about Devosia and Rhodophyticola? They also appear to be "above the 1:1 line".

261: It would be useful to repeat here the duration of these incubations. Or even better, specify them in the figure caption (unless you convert your data into actual rates, then the time

information will be included in the values).

262-263: Inconsistent formulation. "incorporated low background levels" vs. "not significantly different from the killed controls". Furthermore, this statement is true for DON, but not for DOC. In Fig. S4, the right-most panel, there is a sizeable sub-population of algal cells from the axenic culture with a very high ¹³C-DOC uptake, but it appears to be ignored (nothing is mentioned about this feature in the data). By comparison, the distributions of Cnet among cells for the treatments with *Thalassospira* and *Microbacter* look very similar to the cells from the axenic culture, yet they are interpreted differently.

269: Fig S5 A & C: The numbers on the x and y axes do not match the axis labels. The % sign in the labels should be omitted, or the values should be converted to %.

275: "similar" - Difficult to see if the axes do not have the same scale.

277: which data? species means, species medians, values in individual cells?

288: Why only *Marinobacter*?

291-292: The purpose and implications of this statement here are unclear.

296-297: Why not testing here also for glucose assimilation?

299: Which "previous experiments"?

302-304: What about the sub-populations clearly seen in Fig. S7A? They deserve to be mentioned.

306: "isotope labeled nitrate and glycolate were incubated..." - this does not make sense to me.

309: add "growth based on" before "glycolate incorporation"

314-316: If Fig S4 shows very little DON incorporation for *Roseibium* and *Tepidicaulis*, which DON do they require for growth then? Also, *Henriciella* is not included in Fig. S4, so it is hard to see how this statement is justified for this species.

320: Why is *Henriciella* missing in Fig. 1 & S4?

320: Why is also *Alcanivorax* not included in this list? It has a similar range of DON and DOC uptake as, e.g., *Marinobacter*. I can see that this species differs with respect to DOC transfer to the algae, so, perhaps, it belongs to yet another category. However, since it is the only one with this type of interaction, it has not been detected by the cluster analysis. Please discuss briefly this possibility.

322: Unclear how the hypothesis ("these strains are highly efficient at organic matter uptake") follows from the evidence mentioned in the previous sentence (low DOM uptake).

329: "generally matched". What do you have to say about the exceptions?

330-332: The purpose and implications of this statement (evidence) are unclear. Do you suggest that these species grow on DIC? Also, how do you interpret the negative % values in Fig. S8?

356-357: add "a genetic potential for" after "presense of". This statement is very important and should be more emphasized.

359: This section title, when read as a statement, seems overstated. It does not seem to be supported by evidence.

362: You identified 3 "larger" strategies, so why not testing also proteome of a representative

from the 3rd category?

381-383: This is interesting, but it is not quite clear why it is mentioned here. Similar for I. 386-387.

394-396: Yes, but the C and N uptake by these strains is still very low compared to the others. Thus, the usefulness of this information is unclear.

404-410: I agree that this is a useful conceptual categorization of the interactions. But, first, what are the boundaries separating the categories? Second, it would be useful to describe the categories more clearly and thoroughly, i.e., by saying what they do in addition to what they don't do (e.g., "macromolecule users remineralize C, but not N, to the algal cells"; "small-molecule users did not incorporate DON but also did not remineralize DON"). Also, it is important to emphasize here that this categorization is specific for the studied diatom species and experimental conditions (essentially F/2-media type of environment, probably also large cell densities).

408: For the small-molecule users, some of them assimilate nitrate, but not all of them (Tepidicaulis, Roseibium). What is, then, the likely N source for the later ones?

409: "highly efficient recyclers" - except for Henriciella?

411: Do you mean "wide cell-to-cell variation"?

422-423: Another reason why Alcanivorax could be considered yet another category. And similar to Pusillimonas. But, as mentioned earlier, introducing any new category faces the question about its boundary.

437: This is the only place where the high nitrate availability is mentioned, in a rather unnoticeable manner. This point needs to be more strongly emphasized.

442: If they did not use nitrate for growth, what was their N source then?

456-457: As mentioned earlier, I agree with this suggestion, but the question remains: how would it be done in practice, considering that real systems contain many more algal and bacterial species? Normally, genomic/proteomic data is collected from communities, not activity data.

457-460: This suggestion seems to be somewhat over-stretched. The present data suggest that if a specific algal species releases DON and this DON is remineralized by algae-associated bacteria, the form of this remineralized N may not necessarily be effectively taken up by the same algal species (*P. tricornutum* in this case). However, it can be taken up by another one. So there is still a long way to go to make this categorization practical for models of complex phytoplankton-associated microbial communities.

465-466: DOC/DOM users post-cultivation. Unclear why this type and at this phase. Please explain better the logic.

Table 1: *Henriciella* does not seem to appear in some graphs. Add units to cell abundances.

615: Are cell counts available in addition to the Chl a fluorescence data?

625: It should be emphasized in the caption that these data originate from the *same* SIP experiment, just the measured and analyzed cells are different (bacteria, A, vs. alga, B). During the first reading of the figure I got somewhat confused about this.

629: What do you mean by "single copy genes"?

633: Explain PATRIC

Figure 1: The graphs need to be improved (technically and esthetically). There are no x & y-ticks,

the "black" class in (A) seems to be annotated upside-down. The black points could better be colored as well. There is no need to annotate the "minus" part of the axis in (B).

Figure 2: I do not see the nanoSIMS color bar to be applicable to the colored boxes shown next to "DOC to bacteria", "C to algae", etc. For example, the dark-cyan or darkish-blue color is pretty abundant in those boxes, but there is no such color in this color bar. Please print the figure out to verify the colors.

Reviewer #2 (Remarks to the Author):

In the submitted manuscript, "Single cell carbon and nitrogen remineralization profiles...", the authors Mayali et al. present a thoughtful, robust, well-designed, and statistically supported experimental examination of bacterial-algal interactions using cutting edge nanoSIMS analysis. This work is transformative for the field of microbial ecology as well as for industrial-scale algal culturing. In the past, the field has largely focused on correlative "interactions" between bacteria and algae, but mechanistic examinations of trophic interactions and population-resolved activity are desperately needed in order to properly link microbial communities with both environmental biogeochemistry and industrial products. Not only do the authors provide quantitative measurements of C and N uptake driven by the interactions between bacterial taxa and the model organism *P. tricornutum*, they also provide a new conceptual framework for categorizing these interactions. The conceptual framework and functional categories revealed here help to further illuminate the bacterial "black box" and may be useful for the incorporation of bacterial interactions in biogeochemical models in the future.

One conclusion highlighted by the authors in the abstract that is not fully supported by their analysis is the claim that the "functional categories were not linked to phylogeny and could not be elucidated strictly from metabolic capacity as predicted by comparative genomics."

It is perhaps unsurprising that the authors were unable to link the presence and absence of functional pathways with phylogenetic placement. Each functional pathway examined is likely conserved at variable phylogenetic levels, ranging from deeply conserved (i.e., at the Family or Genus level) to more recently acquired (i.e., species/sub-species/"microdiverse" level) traits (see Martiny et al 2015, Science, for more details). However, it is surprising that the authors could not find any links between the presences/absence of either carbon or nitrogen pathways and the bacterial incorporation/remineralization of algal C/N. The authors rightfully point out that the presence and absence of the pathways in their bacterial genomes should be interpreted with caution. Nevertheless, it would be useful to verify their conclusions with an additional clustering analysis. Specifically, if the bacterial genomes are clustered based on (1) the presence/absence of C pathways, (2) the presence/absence of N pathways, and (3) the presence/absence of all pathways, are the authors able to recreate clusters similar to those identified by their Kmeans clustering analysis? If not, then their original conclusion will be better supported. However, if they are able to recreate their Kmeans clusters, then this analysis would suggest that there is a link between genome content and metabolic interactions between bacteria and *P. tricornutum*.

Overall, this is a well-written and compelling analysis of single cell microbial interactions with important implications for global elemental cycling that will be of significant interest to the readership of Nature Communications.

Minor Comments:

Line 83: Delete ";"

Line 97-100: Run-on sentence, separate into two.

Line 111-115: Run-on sentence, separate into two.

Below we address reviewer comments **in bold**.

Reviewer #1 (Remarks to the Author):

The authors present an intricate and extensive dataset obtained via a series of isotope-labeling incubation experiments involving one diatom species (*Phaeodactylum tricornutum*) and 15 diatom-associated bacterial species. The results include data on C and N assimilation obtained with a single-cell resolution (by nanoSIMS) as well as genomic, proteomic, fluorescence, GC-MS, and cell count data obtained on bulk samples. Based on this dataset, they propose functional categorization of the diatom-associated bacteria into "macromolecule remineralizers", "macromolecule users", and "small-molecule users". They also suggest that this categorization can, ultimately, be useful in mechanistic models (conceptual but also numerical) of C and N cycles in aquatic environments.

Overall, I find both the experimental design and data interpretation to be sound and well executed. Especially the number of analyses, and their diversity, are very impressive. Also, the final conceptual synthesis of the results, as proposed by the authors, is useful outside the scope of this study. However, as outlined below, I feel there are quite a few aspects that need to be clarified before the manuscript can be considered for publication.

1. The categorization is based, essentially, on the single-cell ¹³C and ¹⁵N data obtained by nanoSIMS. The power of this approach is that it reveals the pronounced cell-to-cell heterogeneity within populations that are typically considered as "homogeneous". This is well demonstrated by the present data, which shows that although a group of cells is thought to be the same in one respect (e.g., because they belong to a specific bacterial species or have a specific genomic make-up), the activity (quantified through ¹³C and ¹⁵N assimilation) substantially differs among the individuals. This raises a larger question: how useful is the categorization into groups, if individuals (and likely also many individuals rather than just a few) from one (functional) category behave, or appear to behave during a given experiment, as members of another (functional) category?

We appreciate the reviewer's thoughts on this matter. We do believe that it is indeed useful to categorize different bacterial taxa into functional categories, even if the high cell-to-cell variability results in overlap across categories. In our experiments, even in the absence of cell-to-cell variability within one culture, we would expect overlap in activity across strains because the growth substrates are complex (algal organic matter includes a very high chemical diversity of molecules). So overlap across taxa does not necessarily mean those taxa are growing on the same substrates. Based on this reviewer's suggestion, we have now more directly quantified cell-to-cell variability by calculating coefficient of dispersion (CD), which uses the quartile values (Q1 = 25%, Q2 = 50%, Q3 = 75%) to quantify dispersion: $CD = (Q3 - Q1) / (Q3 + Q1)$. We did not use coefficient of variation since most of the data were not normally distributed. Using this CD calculation, we now show that higher variability in organic N incorporation by the bacterial cells is correlated to increased N remineralization to the algal cells. We did not find such a connection between high heterogeneity in C incorporation associated with higher remineralization. We greatly appreciate the reviewer suggesting a further analysis on the cell-to-cell variability as it led to us finding this

connection.

I am bringing up this issue because I feel that the authors should touch upon intercellular heterogeneity revealed in the nanoSIMS data, but they don't. There are several issues to address.

1a. Why is there such a large intercellular heterogeneity within one bacterial species (Fig. S4A & S7A) in the first place? The same question applies to the diatom species (Fig. S4B).

Indeed, we and others who have documented such single cell variability in cultures (in this manuscript and previous publications) report it, usually as a median value since the data are generally not normally distributed, but then that variability is not really taken into account in the downstream interpretation. As mentioned above, we have now calculated a coefficient of dispersion. In this particular dataset, we attribute the cell-to-cell variability in isotope incorporation by bacteria and diatom cells to the following reasons (in no particular order of importance, because we do not know which factor is responsible for which fraction of the variability; we have added this information to the manuscript):

- 1) The cultures were in batch mode and not in chemostats, and in addition were not synchronized, so cells (both algae and bacteria) are in different phases of their division**
- 2) The cultures are not agitated, and *P. tricornutum* exists both as a benthic and pelagic organism, often forming aggregates of multiple cells, so there is going to be spatial heterogeneity of resources**
- 3) Bacterial cells can be either algal-attached or free-living, which would make them likely to have different accessibility to external added labeled algal organic matter, again leading to spatial heterogeneity of resources. Even among free-living bacteria, they are incorporating the labeled extracellular algal OM and unlabeled OM being exuded from the algal cells, and the latter should not be homogeneously distributed.**

Regardless of the reasons for the variability, it is clear that some of the strains are more variable than others. Our calculations of CD (coefficient of dispersion) now show this. We have added a short paragraph to discuss this variability. Other single-cell nanoSIMS publications have addressed this variability in the context of adaptation to a changing environment (see Zimmermann et al 2015, Schreiber et al 2016, Calabrese et al 2019), thus we kept this discussion relatively short as the examination of the variability was not the aim of our study (though we agree with the reviewer that we should have discussed it further than we did initially).

1b. Was the clustering based on Kmeans derived from ^{13}C and ^{15}N data of **all** individual cells, or only using the median values derived for each species? Figure S5-B suggests the latter, but I wonder whether this is a correct approach. It would be useful to see how the different individual bacterial cells fall (or not) within the depicted cluster boundaries. It also raises a bigger question related to the one above: how well defined, and thus meaningful, **are** the boundaries?

- Indeed, the Kmeans clustering used the median values. We did this because we wanted to use the bacterial C and N incorporation data and the algal C and N incorporation data (so all 4 variables together). If we were to use the single cell data, we would have to do the

analyses separately for the bacterial cells (using 2 variables: bacterial C and N incorporation), and then for the algal cells (2 variables: algal C and N incorporation). The one way to potentially do such an analysis would be if all or most of the bacterial cells were attached to the diatom cells (they were not), and then we could link individual bacterial cells to specific diatom cells, and each algal-bacterial pair would have 4 variables. Unfortunately, the great majority of the bacterial cells were free-living (even for the strains that occasionally attach), and such an analysis was not possible.

1c. Is the median value the right parameter to use when comparing the nanoSIMS data with other data? For example, genomic and proteomic data are done on bulk cell biomass, so represent an "average cell". But the median may not be a good representation of the average cell if the distribution of the ^{13}C or ^{15}N enrichment is skewed (which it seems to be for most species) or shows a clear division into two (or perhaps more) sub-populations (Fig. S4A, S7A).

Indeed, the issue here is the comparison of a bulk measurement versus a set of single cell measurements. The first step was for us to at least attempt to do this, and thus we needed a metric to summarize the single cell data, in order to also carry out tests of significant isotope incorporation. The single cell isotope incorporation data were not normally distributed and thus we reported median values and carried out non-parametric statistical analyses to test for significant isotope incorporation compared to killed controls or axenic cultures. Just to make sure that using medians did not skew our results, we statistically compared means and medians for the 4 datasets (C and N incorporation by bacterial cells and C and N incorporation by algal cells). Those 4 linear regression analyses yielded R^2 values of 0.955 to 0.99 with p-values < 0.0001 , so using means would not alter the patterns. Note that medians were either equal to (slope of 1, for bacterial data) or slightly lower than mean values (slope less than 1, for algal data).

The second comparison was to examine the single cell data in the context of other bulk data, primarily genome content. We do not expect any cell to cell heterogeneity in genome content since these isolates are clonal and are unlikely to have had the time or the selective pressure to diversify their genomic content. If we had designed our experiments to compare nanoSIMS data to proteomics or RNA expression data, we agree that since we would expect differences in single cell expression—it would make sense to try and look at the median of just the active subpopulations in the nanoSIMS data which could be more equivalent to the average protein expression. Such a calculation would involve removing the non-active cells, and recalculating a metric of isotope incorporation (likely median, unless the data without zeros are now normally distributed).

1d. This inconsistency between the mean and median values implies that the comparison of genomic/proteomic data with nanoSIMS data may not be as meaningful as we would like it to be, especially if we do not understand the reason behind the large intercellular heterogeneity seen in the nanoSIMS data and because we do not have the corresponding genomic/proteomic data for each individual cell that was measured by nanoSIMS (this is a more general problem, not specific to this study). To put it more bluntly, it is unclear what we are categorizing here, and why it is meaningful: individual cells, or individual cells that have already been "pre-categorized" (e.g., by assigning them to species, or by calculating a median for a population)?

Since means and medians were similar to one another and exhibited strong and statistically significant correlations, we believe this comment has been addressed. Also, we now

incorporate the cell-to-cell heterogeneity in ^{15}N incorporation which maps well to the categorization of macromolecule remineralizers vs users.

2. The authors chose to search for links between phylogeny (as derived from 16S rRNA) and activity (or a type of ecological interaction derived from the activity), but found none. However, the same conclusion can be made for the larger genomic (Fig. 2, squared diagrams at the bottom), or even proteomic (Fig. 3), dataset when compared with the nanoSIMS dataset. We simply cannot predict from the genomic potential, or even the proteomic make-up, of an *average* cell how active an *individual* cell will be in terms of C and N assimilation. I think this gap should be more emphasized by the authors (they mention it gently in the abstract, l. 31-32, but I think it deserved a bigger discussion).

We fully agree with this statement. In the original manuscript, we wanted to be careful not to be too controversial and indeed we mentioned it “gently”. We have sometimes strengthened our statements, but also removed this concept from the title as suggested by this reviewer.

3. Given the exhaustive analysis of cells by nanoSIMS and genomics (15 species), the proteomic analysis on 2 bacterial species feels somewhat unsatisfying. I understand it's a lot of extra work, but tremendous amount of work has been put into this study already. The question is, however, whether we could actually learn something from the proteomics data in the context of the aim of this study. For example, would the proteomic data be sufficiently strong to allow prediction of the ecological interaction between the diatom and a given bacterial species (i.e., assignment to one of the three categories defined by the authors)? Looking at the proteomic profiles for the 2 species shown in Fig. 3, I am not sure. Actually, the usefulness of the proteomic is not quite clear to me given the fact that 3 functional categories have been identified by the authors, whereas representatives from only 2 categories were studied in greater detail.

We appreciate this comment. We wondered ourselves whether doing proteomics on all the strains rather than on just two would help link activity and genomic function. This seems unlikely, especially given the above concern about average proteomics activity and single cell nanoSIMS measurements, and the likelihood that the important protein expression of active subpopulations would be lost in the noise. The aim of the proteomics was not so much to do that, but rather to compare a bacterial strain that grows abundantly in the presence of the diatom (*Rhodophytocola*) and one that does not (*Marinobacter*). We have added some text to address this comment.

===

I provide the rest of my comments as they come to my mind, which roughly follows the main text (line numbers indicated), because ordering based on priorities (more vs. less important) would be too exhausting.

1-2: The title makes too broad a statement while the results are limited to one diatom species, and even under specific culturing conditions (e.g., in an F/2 medium). Given point 2 above, I don't find it useful to emphasize the uncoupling to phylogeny in the title. In fact, the authors

should clarify why the existence of a link between phylogeny and the magnitude of the diatom-bacteria exchange of C and N would be a viable hypothesis (to me, it is not). It would be better to make the title more neutral and to the point of the study, something like "Single cell carbon and nitrogen incorporation and remineralization profiles of bacteria associated with the diatom *Phaeodactylum tricornutum*", or perhaps something shorter.

In order to highlight the concept of using activity data to assign bacteria to functional categories, we have changed the title to “Single cell isotope tracing reveals functional guilds of bacteria associated with the diatom *Phaeodactylum tricornutum*”

32-35: Yes, but the data is insufficient to generalize this statement (only 2 species compared). Also, looking at the overall proteomic profiles, it seems difficult to see that they could be used to predict the categorization of the bacterial strains.

Due to the word limit on the abstract, we have removed this statement

35-37: I can fully identify myself with this statement. But how will the categorization be done to make it practical in situations mentioned in the subsequent sentence (i.e., for models)? For now, only genomic (and perhaps proteomic) based categorization of (natural=mixed) microbial communities can be done with a sufficiently high throughput and sensitivity/resolution. The authors basically show that those type of results are more or less "useless" if we are to understand the ecological interactions within the community in terms of C and N fluxes - because they cannot predict categorization even into the three major functional categories identified by the authors. In other words, the proposed categorization may be useful for modellers, yes (1.38-39), but how will they constrain them based on practically available data? **Indeed, the reviewer makes a great point here. We suggest that it is possible to link activity and genomic data, but that this will require identifying the unknown genomic pathways that result in these activity categories (perhaps requiring single cell RNA or protein expression, and more exhaustive RNA/protein expression experiments coupled with activity measurements under different conditions). Single cell heterogeneity is also likely to have genomic underpinnings, and this could be another focus of targeted research. Since the abstract is limited to 150 words, we have removed this, but bring it up later in the discussion.**

50-53, 57, 72-73: C and N fluxes is not quite what the authors are presenting here; they present the relative amount of assimilated C and N, as % of the cellular C and N content. Thus, these two formulations prime the reader with somewhat "false" expectations about what is done in this study. I suggest to reword these parts to better reflect the content and focus of the present study.
reworded

54: as written, the sentence does not make sense.
amended

95: Not only identity, but also the larger genetic potential of the studied bacterial species.
amended

99: more accurately, not what is "made available", but what is actually assimilated by (transferred to) the algal cell.

amended

101: be more clear about "representative". In which sense?

amended

103-105: The authors formulate these hypotheses as a main goal of the study, but do not make firm statements about their validity. The correlation hypothesis for C transfer is confirmed, but presented in a rather hidden way (as Fig. S5-C and one sentence). No obvious statement is made about the correlation hypothesis for N transfer.

Initially, we had fairly simple hypotheses about bacterial C incorporation being correlated to bacterial remineralization, and the same for N. Thus we wrote that these were our initial hypotheses. What we found was that the correlations were not particularly strong (especially for N, where there appeared to be a threshold), which is why we put those correlations in the supplemental figures and instead put the figures of C vs N (for bacteria and for algae), which more clearly distinguished the 3 functional groups. We have edited this section by removing those two hypotheses and we clarified and shortened the last paragraph of the introduction (it was a bit too long anyway).

107: not "metabolic function" but "metabolic potential".

amended

110-111: I am not sure how to read this sentence (especially the part after the comma, including the comma).

Amended (we removed "with no external sources of organic C", since it is redundant).

113: I am not quite sure how the labeling of the C and N sources was derived/measured from this experimental design (addition of isotope-labeled *P. tricornutum*-derived organic matter to these co-cultures). How was the isotope-labeled *P. tricornutum*-derived organic matter prepared is also not described.

The procedure to generate the isotope labeled material is described in detail in the methods, but to help the reader here, we added "chemically extracted", we hope this helps.

119: "These" data... Which data? All together, or only those mentioned in the previous sentence?

Amended (it was all together the data, not just the data from the previous sentence).

121: I think the use of "showing" here overstretches the applicability of the results. Especially because the study only focuses on 1 algal species under specific "environmental" conditions (e.g., high nitrate availability). I would agree that the three categories are a useful suggestion for conceptual models, a better first approximation of algal-bacteria interactions than the one currently commonly used.

We replaced "showing" with "suggesting", and added "in our experiments"

128-133: Very long sentence, difficult to follow. Consider splitting to make it easier to follow.

amended

143: 75% - percentage of what exactly?

amended

150: unclear, too brief. transferred where?

amended

155: add hr to the light/dark cycle.

amended

156: Please be explicit about the nitrate concentration here, as it is not immediately obvious - at least to a reader without practical knowledge of cell culturing - that F/2 medium is rich in nitrate. It will be important later on when you mention that the conclusions in this study are relevant when nitrate is available as a source of N.

amended

157-159: Is this describing what is shown in Sup Fig S2? If so, it is unclear how the ANOVA was done, and which "multiple tests" it was "adjusted for". Specifically, what exactly is compared in Fig S2? The last time points, or all time points? When I look at it, the curves all look rather similar and of two rough "types" (similar to either *Marinobacter* or *Devosia*). Thus, it is not clear why the curves shown in the lower-right panel show no significant influence of the bacteria on the diatom growth, while the other panels do show significant influence.

Yes, this referred to Fig. S2. We compared the chlorophyll fluorescence at day 18 across treatments. This section has been clarified.

163: Add "assimilation and" before "remineralization".

amended

165: Add SPE after "solid phase extraction", otherwise the abbreviation is unclear later in the text. Also, clarify how the labeled biomass was obtained in the first place (with a little more detail than just with a citation).

Amended. Detailed protocol is included in the supplementary information, now referenced in the text.

168: add "hr" for the light/dark cycle.

amended

171: You determined the labelling of the added exudates, but what was the labelling of the exudates in the actual incubation? Why wasn't that measured instead? If it were, the assumption about the dilution of the added exudates by the exudates present in the co-cultures would be unnecessary. The information about the substrate labeling is required when calculating the Cnet and Nnet later on. The same for the second experiment (l. 175-177).

Measuring the labeling of the exudates in the actual incubations would indeed have been an improved approach to measuring the labeling of the added exudate. However, this would have involved an additional step: first, solid phase extraction of the labeled spent medium

from the axenic culture, then adding this exudate to the unlabeled culture (for the experiments), and then collecting the spent medium of those cultures at time zero, and a second solid phase extraction, followed by measurement of that extract by NanoSIMS. We did not follow this procedure, but agree that it is preferable for future experiments. For the second experiment with the labeled nitrate, we would need to extract the nitrate from the medium and measure its isotope labeling with gas chromatography, and we currently do not have access to this measurement capability.

178: Why only 11 and not all 15? What determined the choice?

The 4 strains not tested for this experiment were found have been contaminated with Marinobacter which can grow in F/2 medium. This was discovered after 16S Sanger sequencing showed either a pure Marinobacter or a mixture of two of more strains. This has been amended.

181-184: Calculations not clearly described. I am familiar with your previous work, so I can guess what you did here, but for an average reader this description is insufficient. A reference to a previous calculation approach should at least be added, but preferably a more detailed explanation. Of particular importance is to be clear about the labeling of the substrates.

We have added more detailed information to the supplementary material and made a few edits to the main manuscript, we hope this is now more clear.

At this point, I also wonder why you only convert the enrichment data into the relative amounts of assimilated C and N (C_{net} and N_{net} in %) and not to relative rates (%C or %N per day) or even cell-specific rates (mol C cell⁻¹ day⁻¹ or mol N cell⁻¹ day⁻¹). Using the rates would be more appropriate considering that C and N *fluxes* (which are expressed per unit time) are your stated aim of the study. Additionally, relative rates are useful from the perspective of the bacteria (they reflect cell's growth rate on a specific substrate, which is added as a labelled tracer during the SIP experiment), but a cell-specific rate is the relevant quantity when describing an ecological interaction (C or N flux) between algae and bacteria in models. Since this is the context and aim of your study, it may be even necessary that you determine the cellular C and N contents of the bacteria (and diatom) and convert your nanoSIMS data into cell-specific rates. In this way, the differences between the studied bacterial species may decrease or become more pronounced, depending on the differences in their cellular C and N contents and C:N ratios. I wonder whether the categorization of bacteria would differ if cell-specific rates were used instead of the relative amounts of assimilated C and N (C_{net} and N_{net}). Considering these comments, please include a short discussion about your preferred choice of nanoSIMS-derived quantities as a measure of ecological interactions between bacteria and algae.

Thank you for this comment. Since the algal OM experiments were carried out over 24 hours, the C_{net} and N_{net}% are per day, and we have altered some of the language to reflect that. Regarding calculating cell-specific rates based on cell C and N content, we can estimate C and N content based on the 2 dimensional cell size obtained from the size of the regions of interest, and based on the literature, we can assume the bacteria have similar C/N (see <https://journals.asm.org/doi/10.1128/AEM.64.9.3352-3358.1998> which shows that coastal and oceanic marine bacteria have the same C/N). Using this estimate combined with the cell abundances, we have calculated a total C and N flux from algal DON and DOC into bacterial C and N, and have added it to Table 1.

185-186: Please explain more clearly. How does 66% labeled 33% unlabeled "turn" into 50%? It is not clear to me what you are talking about here.

We apologize for not being clear (it's a bit confusing). We added the equivalent of 100% of the DOM at stationary phase, but this was diluted by 50% unlabeled DOM that has accumulated during exponential phase. Thus, 50% unlabeled, and 100% labeled add up to 33% unlabeled and 66% labeled. We simplified this statement, saying the added labeled DOM was diluted by unlabeled DOM equivalent to 50% of the DOM added.

190: calculated -> assumed

amended

192: Unclear. 300 mM is >> 500 uM glucolate added.

This was a typo, should have read 300 micromolar (not millimolar); typo fixed.

193: "to compare across treatment". To compare what?

Amended (we compared algal incorporation to axenic cultures, and bacteria incorporation to killed controls)

195-197: Was the K-means clustering done on nanoSIMS data of individual cells, or using the median (or mean) value for each species? If the latter, how was the cell-to-cell heterogeneity, which differed greatly among the different species, accounted for?

See response at the beginning of this thread. K-means done on median data, and coefficient of dispersion now calculated to address cell-to-cell heterogeneity

206: Unclear why only these 2 species were selected for this analysis.

Explained briefly (also see earlier response on this topic).

220-223: Unclear how this statement is supported by data shown in Fig. S2. I don't know where to look. As mentioned earlier, all curves look very similar to me and of two types.

Rephrased from "growth" to "yield". Indeed, growth rates were not different, just yield (as measured by chlorophyll fluorescence). Note that citation #35 previously reported those same co-cultures also increased chlorophyll yield.

230: replace to "could not be"

amended

231: It would be useful to add here also the algal cell counts. Note that Table 1 does not specify the units of the cell counts.

We did not count algal cells from these experiments. Units added to Table 1 for bacterial cells (these were recounted, this time with flow cytometry).

242: Yes, high variability among isolates. However, there is also high variability among individual cells of a given isolate, and this is not touched upon at all. Can we understand variability among isolates if we do not fully understand variability within isolates?

Here we include some text about our new calculations of the coefficient of dispersion

249: not Devosia?

Nice catch, Devosia should have been included

253: I assume you are talking here about a correlation at the level of median values for each species. But was there also correlation on the level of individual cells? That is, were Cnet and Nnet in cells correlated for a given species, and consistently so for all species?

Indeed, we were examining the correlation with the median values. We also now add a brief statement saying the correlation was also positive and significant at the single cell level.

259: What about Devosia and Rhodophyticola? They also appear to be "above the 1:1 line".

Indeed, they are above the line, this has been amended (note we renamed the line "stoichiometric uptake")

261: It would be useful to repeat here the duration of these incubations. Or even better, specify them in the figure caption (unless you convert your data into actual rates, then the time information will be included in the values).

Amended (added 24 hours to that sentence, reported daily Nnet and daily Cnet in the paragraph)

262-263: Inconsistent formulation. "incorporated low background levels" vs. "not significantly different from the killed controls". Furthermore, this statement is true for DON, but not for DOC. In Fig. S4, the right-most panel, there is a sizeable sub-population of algal cells from the axenic culture with a very high ¹³C-DOC uptake, but it appears to be ignored (nothing is mentioned about this feature in the data). By comparison, the distributions of Cnet among cells for the treatments with *Thalassospira* and *Microbacter* look very similar to the cells from the axenic culture, yet they are interpreted differently.

We have reworded this section. The results are written based only on statistical significance testing. Indeed there are a small number of algal cells in the axenic treatment showing enrichment higher than the killed control, but as a population, the axenic cells were not statistically enriched compared to the killed control.

269: Fig S5 A & C: The numbers on the x and y axes do not match the axis labels. The % sign in the labels should be omitted, or the values should be converted to %.

amended

275: "similar" - Difficult to see if the axes do not have the same scale.

Rescaled y axis in figure so that Cnet and Nnet match, and reworded "similar" to "of the same magnitude"

277: which data? species means, species medians, values in individual cells?

Added median

288: Why only *Marinobacter*?

The Marinobacter co-culture has consistently been the most mutualistic co-culture among all the strains, and is commonly found to be in high abundance in co-culture with microalgae. We added a statement to summarize this.

291-292: The purpose and implications of this statement here are unclear.

We have tried to rephrase, as here we attempted to interpret the GC-MS data to confirm that Phaeodactylum-associated bacteria in co-culture lead to decreased concentrations of algal released organic carbon. Since glucose and glycolate were in lower abundance in the exudate of the co-culture compared to the axenic Phaeodactylum culture, it suggests that Marinobacter metabolized glucose and glycolate. The other alternative is that the presence of Marinobacter caused Phaeodactylum to produce less glucose and glycolate. We believe the former is more likely, though both could occur simultaneously.

296-297: Why not testing here also for glucose assimilation?

We chose glycolate instead of glucose since it seems more relevant to systems where high abundances of microalgae lead to high concentrations of oxygen that lead to photorespiration and thus the production of glycolate.

299: Which "previous experiments"?

Confusing, we have removed this (we meant the previous experiment where we quantified that bacteria incorporate organic N from *P. tricornutum*)

302-304: What about the sub-populations clearly seen in Fig. S7A? They deserve to be mentioned.

We have added a statement that Thalassospira, Stappia and Alcanivorax, in particular, showed pretty dramatic subpopulation, each with some cells with no detectable incorporation, and other cells with relatively high incorporation, which could be a sign of resource limitation. We thank the reviewer for pointing this out.

306: "isotope labeled nitrate and glycolate were incubated..." - this does not make sense to me. **We have reworded this section.**

309: add "growth based on" before "glycolate incorporation"
amended

314-316: If Fig S4 shows very little DON incorporation for Roseibium and Tepidicaulis, which DON do they require for growth then? Also, Henriciella is not included in Fig. S4, so it is hard to see how this statement is justified for this species.

Figure S4 shows incorporation of complex DON that was collected by the PPL solid phase extraction column, whereas the unlabeled DON in the nitrate/glycolate experiment includes all algal-released DON. Thus Roseibium and Tepidicaulis likely incorporate DON that is not captured by the PPL column. We have clarified this statement. Regarding Henriciella, indeed it is not included in Fig. S4 (see answer to next comment below), but since we were able to test nitrate incorporation and it was not detected, we are keeping Henriciella in this sentence.

320: Why is *Henriciella* missing in Fig. 1 & S4?

Henriciella is at such low abundance in the co-culture that its signal is obscured by *Phaeodactylum* cells, thus we could not collect any data when both cell types were collected on 0.2 micron filters. In the glycolate/nitrate experiment, we could detect *Henriciella* cells because *Phaeodactylum* cells were removed prior to incubation. Hopefully this has been clarified with the edits.

320: Why is also *Alcanivorax* not included in this list? It has a similar range of DON and DOC uptake as, e.g., *Marinobacter*. I can see that this species differs with respect to DOC transfer to the algae, so, perhaps, It belongs to yet another category. However, since it is the only one with this type of interaction, it has not been detected by the cluster analysis. Please discuss briefly this possibility.

We added *Alcanivorax* to this list. Indeed this strain is a bit odd in that it did not incorporate DON but remineralized DOC at one of the highest levels. We added a sentence here.

322: Unclear how the hypothesis (“these strains are highly efficient at organic matter uptake”) follows from the evidence mentioned in the previous sentence (low DOM uptake).

We have reworded this. What we meant was that perhaps these bacteria are partially growing on background organic matter that does not originate from the algal cells.

329: “generally matched”. What do you have to say about the exceptions?

Information added

330-332: The purpose and implications of this statement (evidence) are unclear. Do you suggest that these species grow on DIC? Also, how do you interpret the negative % values in Fig. S8?

We apologize for the lack of clarity and we have tried to clarify. The idea is that anapleurotic C fixation is a sign that the cells are growing and are efficient at recycling lost C in the TCA cycle. Thus, when we see incorporation in media with no algal organic matter, we consider this as evidence for growth (or at least metabolic activity) using non-algal OM. Regarding the negative % C values in Fig. S8, we had calculated those values using the standard isotopic ratio of 1.1237%. We have recalculated these % C values, instead using the isotope measurement of the filter onto which the cells were located (the values are now much closer to zero).

356-357: add “a genetic potential for” after “presense of”. This statement is very important and should be more emphasized.

Amended

359: This section title, when read as a statement, seems overstated. It does not seem to be supported by evidence.

We have toned down this section title to “Protein expression hints at bacterial ecological strategy”

362: You identified 3 “larger” strategies, so why not testing also proteome of a representative from the 3rd category?

We carried out these experiments comparing Rhodophyticola and Marinobacter proteomes prior to the K-means clustering identifying 3 clusters, and we chose Marinobacter because we also had collected the GC-MS data from this co-culture.

381-383: This is interesting, but it is not quite clear why it is mentioned here. Similar for l. 386-387.

We believe the reader will be interested to know that CO metabolism genes and stress proteins are highly expressed by bacteria in co-culture with Phaeodactylum, but since it does not fit with the isotope incorporation data, we have moved this to the supplementary.

394-396: Yes, but the C and N uptake by these strains is still very low compared to the others. Thus, the usefulness of this information is unclear.

We have added a statement reminding the reader that the PPL column extraction does not retain some small molecules such as amino acids and organic acids (e.g. acetate)

404-410: I agree that this is a useful conceptual categorization of the interactions. But, first, what are the boundaries separating the categories? Second, it would be useful to describe the categories more clearly and thoroughly, i.e., by saying what they do in addition to what they don't do (e.g., "macromolecule users remineralize C, but not N, to the algal cells"; "small-molecule users did not incorporate DON but also did not remineralize DON"). Also, it is important to emphasize here that this categorization is specific for the studied diatom species and experimental conditions (essentially F/2-media type of environment, probably also large cell densities).

We have amended this section, as perhaps it was not clear that the categorization is based on the K-means clustering shown in Fig S5, where the boundaries are shown in 2 dimensions (perhaps we are not understanding what the reviewer is asking here about the boundaries of the categories). We also had our graphics department improve figure 4.

408: For the small-molecule users, some of them assimilate nitrate, but not all of them (Tepidicaulis, Roseibium). What is, then, the likely N source for the later ones?

We believe the likely N source for these taxa are small molecules not captured by the solid phase extraction protocol. We have added this information.

409: "highly efficient recyclers" - except for Henriciella?

Amended, Henriciella is indeed an exception.

411: Do you mean "wide cell-to-cell variation"?

Here we meant that within a category, there was variability across strains (in other words, not all members of a category had the same incorporation pattern). We have tried to clarify this point.

422-423: Another reason why *Alcanivorax* could be considered yet another category. And similar to *Pusillimonas*. But, as mentioned earlier, introducing any new category faces the question about its boundary.

We agree that the current number of categories likely lumps multiple categories together. Note that the K-means clustering did not identify 4 clusters based on these data (it was either 2, 3, or 8 clusters). *Alcanivorax* and *Pusillimonas* were placed in the same K-means category but when the 2nd dimension is examined (in what is now Fig. S6B), they are on the opposite (as pointed out, suggesting they belong to different categories if we had more than 3).

437: This is the only place where the high nitrate availability is mentioned, in a rather unnoticeable manner. This point needs to be more strongly emphasized.

We have added this information in other places throughout the manuscript to remind the reader that our experiments were carried out at relatively high nitrate concentrations, though high density algal raceways are routinely operated at 4X or more of those concentrations (see new reference #43 added).

442: If they did not use nitrate for growth, what was their N source then?

Amended (likely small DON not retained by the extraction protocol)

456-457: As mentioned earlier, I agree with this suggestion, but the question remains: how would it be done in practice, considering that real systems contain many more algal and bacterial species? Normally, genomic/proteomic data is collected from communities, not activity data.

We have added a few sentences here to address this

457-460: This suggestion seems to be somewhat over-stretched. The present data suggest that if a specific algal species releases DON and this DON is remineralized by algae-associated bacteria, the form of this remineralized N may not necessarily be effectively taken up by the same algal species (*P. tricornutum* in this case). However, it can be taken up by another one. So there is still a long way to go to make this categorization practical for models of complex phytoplankton-associated microbial communities.

We have removed this sentence.

465-466: DOC/DOM users post-cultivation. Unclear why this type and at this phase. Please explain better the logic.

We have rephrased this sentence.

Table 1: *Henriciella* does not seem to appear in some graphs. Add units to cell abundances.

Units added. *Henriciella* is not present in some graphs due to the explanation mentioned above and in the manuscript (abundances in co-culture were too low to analyze single cells on filters where both *Phaeodactylum* and the bacteria were present).

615: Are cell counts available in addition to the Chl a fluorescence data?

We have added some new data (and an additional co-author) that includes flow cytometry bacteria cell counts over the course of a batch culture.

625: It should be emphasized in the caption that these data originate from the *same* SIP experiment, just the measured and analyzed cells are different (bacteria, A, vs. alga, B). During the first reading of the figure I got somewhat confused about this.

Thank you for pointing this out. Caption has been amended.

629: What do you mean by "single copy genes"?

Single-copy genes (or single copy core genes) are genes found as a single gene in a set of genomes (i.e. never duplicated within a genome) and are routinely used for phylogenomic studies

633: Explain PATRIC

Amended (PATRIC = NCBI's Pathosystems Resource Integration Center)

Figure 1: The graphs need to be improved (technically and esthetically). There are no x & y-ticks, the "black" class in (A) seems to be annotated upside-down. The black points could better be colored as well. There is no need to annotate the "minus" part of the axis in (B).

Thank you for the suggestions. We have added tick marks, negative part of the axis in B has been removed, blue and green dots have been recolored, labels moved slightly, and part of the charts expanded in a sub-area.

Figure 2: I do not see the nanoSIMS color bar to be applicable to the colored boxes shown next to "DOC to bacteria", "C to algae", etc. For example, the dark-cyan or darkish-blue color is pretty abundant in those boxes, but there is no such color in this color bar. Please print the figure out to verify the colors.

Thank you for pointing this out. We have recolored the color bar to match the boxes (and in this process, we noticed that the color bars for DOC to bacteria and C to algae had been switched in the first submission, this is now fixed)

Reviewer #2 (Remarks to the Author):

In the submitted manuscript, "Single cell carbon and nitrogen remineralization profiles...", the authors Mayali et al. present a thoughtful, robust, well-designed, and statistically supported experimental examination of bacterial-algal interactions using cutting edge nanoSIMS analysis. This work is transformative for the field of microbial ecology as well as for industrial-scale algal culturing. In the past, the field has largely focused on correlative "interactions" between bacteria and algae, but mechanistic examinations of trophic interactions and population-resolved activity are desperately needed in order to properly link microbial communities with both environmental biogeochemistry and industrial products. Not only do the authors provide quantitative measurements of C and N uptake driven by the interactions between bacterial taxa and the model organism *P. tricornutum*, they also provide a new conceptual framework for categorizing these interactions. The conceptual framework and functional categories revealed here help to further illuminate the bacterial "black box" and may be useful for the incorporation of bacterial interactions in biogeochemical models in the future.

We greatly appreciate this nice summary of our manuscript.

One conclusion highlighted by the authors in the abstract that is not fully supported by their analysis is the claim that the “functional categories were not linked to phylogeny and could not be elucidated strictly from metabolic capacity as predicted by comparative genomics.”

It is perhaps unsurprising that the authors were unable to link the presence and absence of functional pathways with phylogenetic placement. Each functional pathway examined is likely conserved at variable phylogenetic levels, ranging from deeply conserved (i.e., at the Family or Genus level) to more recently acquired (i.e., species/sub-species/“microdiverse” level) traits (see Martiny et al 2015, Science, for more details). However, it is surprising that the authors could not find any links between the presences/absence of either carbon or nitrogen pathways and the bacterial incorporation/remineralization of algal C/N. The authors rightfully point out that the presence and absence of the pathways in their bacterial genomes should be interpreted with caution. Nevertheless, it would be useful to verify their conclusions with an additional clustering analysis. Specifically, if the bacterial genomes are clustered based on (1) the presence/absence of C pathways, (2) the presence/absence of N pathways, and (3) the presence/absence of all pathways, are the authors able to recreate clusters similar to those identified by their Kmeans clustering analysis? If not, then their original conclusion will be better supported. However, if they are able to recreate their Kmeans clusters, then this analysis would suggest that there is a link between genome content and metabolic interactions between bacteria and *P. tricornutum*.

We find it fascinating that reviewer #1 suggested that we have a stronger statement about the lack of a link between activity and genomic potential (though asked us to remove it from the title), and reviewer #2 suggests the exact opposite. As suggested by reviewer #2 (many thanks), we carried out additional clustering analyses using C pathway presence/absence, N pathway presence/absence, both C and N pathways, and then all pathways (beyond C and N, including other metabolic pathways). This is now found in new figure S9. We found that the clustering using the genome information does not match the clustering using the activity measurements, which confirms the general lack of congruence between genome and activity.

Overall, this is a well-written and compelling analysis of single cell microbial interactions with important implications for global elemental cycling that will be of significant interest to the readership of Nature Communications.

Minor Comments:

Line 83: Delete “;”

amended

Line 97-100: Run-on sentence, separate into two.

amended

Line 111-115: Run-on sentence, separate into two.

amended

Reviewer #1 (Remarks to the Author):

I appreciate author's careful attention and thoughtful response to my comments and suggestions from the first review round. The manuscript is now clearer, and the conclusions now more accurately reflect the experimental data. Also the extra discussion points add value to the manuscript.

I have additional comments/suggestions that need to be addressed to complete the journey. They are mostly specific, and often minor, and I will write them below based on the line numbers in the updated manuscript (the version without tracked changes).

One major point that requires more attention concerns the observed cell-to-cell variability in the axenic culture of *P. tricornutum* incubated with the algal-derived DOM. Presently, this variability appears to be "brushed over" as a minor detail. My suggestion is that the authors explore and discuss alternative interpretations of this data, because it may represent an "inconvenient truth" that questions the overall interpretation of the data (or at least of the NanoSIMS data), namely that some algal cells seem to be able to do alone the same thing as most of the algal cells can do with the help of bacteria. I provide a more thorough discussion about this issue below, and it would be useful if the authors discussed whether or not (and why not) this could be an alternative interpretation of their data.

Another major point involves the interpretation of the chlorophyll fluorescence data in the context of the impact of bacterial strains on the algal yield. More details below.

I encourage the authors to carefully revise the entire Supplement. It contains a lot of important information, but its presentation feels at times quite sloppy, especially the clarity of some figures and quality of some captions (see specific comments below). For example, the title of the supplement is still the old one.

Additionally, after having gone through the data again, it would be useful if the single-cell Cnet and Nnet data (essentially all data presented in Figures S4 and S7) were made available through a public data repository or as supplementary data (i.e., not just figures) accompanying the manuscript. This is a common practice when publishing NanoSIMS results these days, allowing others to analyze the data from other perspectives, should they want to do that, and facilitating complete transparency of the publishing process.

Specific comments:

L.23: Strictly speaking "bacterial remineralization" cannot be analyzed at a single-cell level with nanoSIMS, as only *assimilated* C or N (by bacteria or the algae) can be quantified. Indeed, for the remineralization part, what is measured and interpreted is the remineralized algal-derived DOM transferred to, and incorporated by, the alga. It would be good to reformulate this sentence accordingly. I tried but could not find a "quick fix". I understand that the meaning of "remineralization" is clearly defined by the authors on L.87-90, but it should also be clear and precise in the abstract.

L.30-31: This sentence is an important outcome of the study, but lacks a conclusive remarks that highlights the take-home message. It could be combined with what was written earlier, something along this idea: "... and could not be elucidated strictly from metabolic capacity as predicted by comparative genomics, highlighting the need for direct activity-based measurements in ecological studies of microbial metabolic interactions."

L.102: Better write "our" instead of "these", because in the preceding sentences you are not referring to any data as such, but rather to methodological approaches used to acquire them.

L.143: Please use something like "regime" after "light/dark" and "prepared" before "using Instant Ocean salts" in this sentence. Also, try to revise this very long and complex sentence, possibly by cutting it into two simpler, more straight-forward sentences.

L.145: "We tested growth impacts" – yes, but please clarify for which purpose and what specifically was used as a measure of "growth". This was an issue earlier, and I think it has not been resolved satisfactorily yet, as I try to explain in the following.

It is explained now that chlorophyll fluorescence was measured and used as a measure of algal biomass. It is also explained that the fluorescence at the last time-point (18 day) was used to test the impact on growth. But this latter choice seems to be rather arbitrary. Why was it not tested for example at day 15, or 13? It would seem more appropriate to use the *entire* time-series (shown in Fig. S2) and explain why the variations of algal fluorescence over time are significantly affected by the bacteria for some of the alga-bacteria co-cultures, and not for some others, although they all show pretty much the same shape (as far as I can see in panel D in comparison to panels A-C). Specifically, the patterns in Fig. S2 seem to group into two groups based on the difference on day 4. From day 7-8 onward, the mean values appear to be systematically higher (although probably not always significantly higher for a given individual time point) for all co-cultures compared to the axenic culture. Yet, the conclusion is that only for 3 out of the 15 co-cultures this effect is significant, which I find quite counter-intuitive.

Also, it can be assumed that the "growth impact" will depend on the number of bacterial cells in the co-culture and their activity. If the activity were the same for the different bacterial strains (which they were not, but suppose now that they were), the impact of a particular strain could be detectable if the corresponding cell counts were larger, whereas the impact could be undetectable if the cell counts were lower. In other words, cells counts of the bacteria in the co-cultures matter probably as much as their activity. But the bacterial cell counts (such as in the initial bacterial inoculum) were not under direct control in these experiments, or were they? If not, how can we conclude anything about the impact of a specific bacterial strain if we do not account for their abundance?

Overall, I am quite confused as to the purpose of the chlorophyll fluorescence assays, and how their results, alone and in combination with the results of the nanoSIP experiments, help us understand the role (impact) of the different bacterial strains.

I think the confusion stems from the unclear description in the "Algal growth effects and microscopy" section. What is meant by "any other experiments" on L. 137? Did you grow the co-cultures for >6 months (>6 transfers, monthly) and then used a subsample from them to conduct the nanoSIP incubation experiments? Similarly, did you use a similar subsample to inoculate 96 well plates with the F/2 medium, and then measured growth of the algae in these plates through monitoring of chlorophyll fluorescence? If this was the case, how was it ensured that the starting point of the co-cultures (algal cell count?) were the same in order to ensure that the time-series (shown in Fig. S2) could be comparable among each other?

Please explain these steps better, otherwise the origin of the co-cultures in each experiment, and the contribution of the "growth-impact" and nanoSIP experiments to the overall understanding of the diatom-bacteria interaction, are not clear. The reader should not be left guessing.

L.155: "was added". As explained in my previous comment, please better clarify the origin and history of the co-cultures just prior to the labeled substrate addition.

L.159: What do the "18" and "45" values mean in atom %? Is this a typo?

L.160: Is Ref. 29 (Dekas et al. 2019) is used to point the reader to the approach used for calculating Cnet and Nnet, or to clarify how to measure ^{13}C and ^{15}N enrichment of the exudate by nanoSIMS after spotting on a Si wafer?

L. 165-169: I find this formulation somewhat confusing, hard to figure out the connection between the 11 strains and the 4 strains in the added sentence. Perhaps it would be clearer if written something like this: "For a third isotope tracing experiment, we tested the anapleurotic activity under extremely low nutrient conditions by incubating the strains in autotrophic medium (artificial seawater ESAW) with no added carbon plus ^{13}C -labeled bicarbonate (1.5 mM added) for one week. Results of these experiments are only available for 11 of the 15 strains because 4 strains

showed signs of contamination.”

L.171: It would be useful to start a new paragraph with “To compare across strains, ...”

L. 175-180: With this text, do you mean that the ¹³C and ¹⁵N atom fraction of the substrate taken up by the cells was assumed and not directly measured? Was it the same in incubations for all co-cultures? For the sake of transparency, it would be better to include more details on this in the supplement.

L.192: It may not matter too much, but it would perhaps make more sense to define CD as a difference between Q1 and Q3 divided by the *average* of Q1 and Q3, so $CD = 2 \cdot (Q3 - Q1) / (Q3 + Q1)$.

L.192-195: Here, you extrapolate from single-cell Cnet and Nnet data to the entire bacterial population. It is unclear, however, why you do this.

Additionally, the population-specific C and N assimilation depends on Cnet and Nnet of individual cells, cell counts, and on the cell size/biovolume (and thus the cellular C and N content). These latter data are, however, not provided, making it impossible to qualify whether the differences in the total DOC and DON daily incorporation among strains (reported in Table 1) are due to differences in Cnet and Nnet, cell counts, or cell size/biovolume. The cell size, biovolume, and C+N content data could be included as an “extended” version of Table 1 in the Supplement along with the Cnet+Nnet values and cell count data.

Last but not least, when describing the process of estimating the total DOC and DON daily incorporation, it would be good to reflect on the findings in this paper, doi.org/10.3389/fmicb.2021.621634 (“Calculation and Interpretation of Substrate Assimilation Rates in Microbial Cells Based on Isotopic Composition Data Obtained by nanoSIMS”), which discusses in quite some detail how to extrapolate single-cell isotope enrichment data to population-specific C or N assimilation rates.

L. 219: “maximum *P. tricornutum* chlorophyll yield”. This is contradicting what you describe in the methods section as a “surrogate for growth”. In methods, you write that you measured chlorophyll fluorescence. I assumed that it is the fluorescence yield measured at very low (actinic) light in dark-adapted cells, which is an accepted way to approximate algal biomass. There is also *variable* chlorophyll fluorescence, which is used to quantify a quantum yield of PSII, which is a proxy for photosynthetic electron transport. In your expression here, you mix “maximum” (which maximum?) with chlorophyll yield (fluorescence or quantum yield?), and additionally use “algal yield” (in Table 1) or just “yield” (in Fig. S2). Together, these ambiguities make everything rather confusing. Please rewrite it more accurately and check for consistency between the main text and supplement.

L.254: “highest incorporator”. As noted above, to which extent is this because of the highest cell counts?

Also, for consistency of notation, it would be better to change in Table 1 the units from microgram DOC and microgram DON to microgram C and microgram N.

L.257: again, it would be useful to clarify to which extent this is due to the high cell counts.

L.260: “...as well as at the single cell level ($p < 0.001$)”. With this add-on, the sentence as a whole does not read well. Please revise.

Also, it is unclear whether this correlation at the single cell level was observed for each bacterial strain separately or when Cnet & Nnet values from all bacterial cells were tested together. I assume the authors report the result for the latter test, but it would also be useful to test the correlations separately for each individual strain to quantify the coupling between the uptake of DOC and DON in the individual bacterial strains (strong and weak coupling would be indicated by high and low R-squared, respectively). I appreciate that this is not the focus of this study, but on

the other hand, the data is there, so it's only a simple step to visualize it in a graph (or multiple graphs, if one graph gets over-crowded) and report R-squared and p-values. This information could be included in a Supplementary Figure and Table, and the corresponding text (probably 1 sentence) with the key outcome could be included at the end of the paragraph (L. 266+). I ask for this because it becomes relevant in the discussion about the sources of cell-to-cell variability in Cnet and Nnet.

L. 266: For completeness, it would be useful to mention that Yoonia and Pusillimonas lie close to the stoichiometric line, as opposed to above or below that line.

L. 268: At this point of reading, it is difficult to recall the experimental design (three separate SIP experiments). Therefore, it would be useful to include a short clause at the beginning of this sentence to indicate, and in fact emphasize, that this algal data (algal Cnet and Nnet) is from the same SIP experiment (incubations) as the bacterial Cnet and Nnet data reported in the preceding paragraph.

L. 270-273: This finding deserves more attention. First, for the sake of completeness, please clarify what you mean by "minority". What percentage of the total number of cells belong to this sub-population?

Second, although cells in this sub-population may be a minority, their ¹³C enrichment (Cnet) is pretty much in the same range (about 0.4 to 1.2 %) as the ¹³C enrichment in algal cells co-cultured with bacteria. This is interesting and can be interpreted in different ways, as I try to outline below.

One possible interpretation is that a fraction of the algal DOC has a form (a specific form, albeit yet unknown) that can readily be re-assimilated by the algae without a need for modification ("remineralization") by bacteria. That only a small fraction of algal cells does this is intriguing and should have a reason. Although I cannot think of one right now, the observation of that sub-population clearly points to significant differences among (presumably genetically equal) algal cells. Perhaps the studied algal species has a "special" metabolic potential to readily assimilate certain forms of DOC, and some of the cells do express this potential randomly, e.g., as a result of stochastic gene expression, during a 24 hr period.

Considering this "background" interaction (axenic alga vs. algal-DOC), the role of bacteria in the co-cultures must then be considered *relative* to this "background". One possibility is that the bacteria modify ("remineralize") the algal DOC such that the remineralized products can be assimilated by the *majority* (essentially all, as seen for most of the co-cultures) of the algal cells, i.e., not only by those that expressed a "special" ability, as suggested above. Intriguingly, the incorporation rate of the "virgin" DOC is very similar to that of the bacteria-remineralized DOC (as far as I can see in Fig. S4-B). Hence, on a single-cell level, it appears that the bacteria do *not* to speed-up the transfer of alga-derived DOC to algal cells, but rather make the alga-derived DOC available to more (all) algal cells. This rises a possibility of another interpretation: that the bacteria do not mineralize the DOC at all, but rather provide a signal to the algal cells that there is plenty of DOC available, which triggers *all* algal cells (as opposed to only those few that engaged in DOC incorporation "randomly" as a result of stochastic gene expression, as suggested above) to switch to a metabolism involving incorporation of the "virgin" DOC. Although this latter possibility seems somewhat stretched, it is consistent with the NanoSIMS data presented in Fig. S4-B.

My point here is that the authors should explore alternative interpretations of their data, such as the one proposed above based on the NanoSIMS data available during this review process. The authors may incorporate the above hypothesis, or refute it if they have the required evidence. In any case, the minimum that the authors need to do is to revise their conclusion that "a minority of the [algal] cells potentially incorporated DOC". It is not true that the cells "potentially" incorporated DOC. Their NanoSIMS data show that those algal cells *did* incorporate DOC (as seen by the distinct ¹³C enrichment of the cells), and they did so rather significantly. It means something, but presently it is presented as a minor detail. I think this detail deserves attention and proper discussion, because it may represent an "inconvenient truth" that questions the overall

interpretation of the data (or at least of the NanoSIMS data alone), namely that some algal cells seem to be able to do alone the same thing as most of the algal cells can do with the help of bacteria.

L.280: I think it would be more accurate to write "incorporation" instead of "metabolism".

L.281: "led to". I think this word suggests a causal relationship, but such a relationship is not supported by the data. A correct formulation would be "was associated with". Also, it would be useful to state that although the relationship was very close to significant, the predicting power of that relationship was very poor (as indicated by the low value of R-squared).

L.282-286: In this sentence, it is unclear that the subject are the algal cells in co-cultures. I agree that the context is clear because the entire text is about co-cultures, but here it would be useful to re-emphasize it by writing something like this: "Regarding N, algal cells in nine co-cultures exhibited significant ¹⁵N incorporation compared to the axenic culture, except when cultured with *Algoriphagus*, ..."

L.290: "method identified three functional guilds". Was this identified "by itself" or because max of 3 groups were allowed during the cluster analysis? Maybe it is mentioned in the methods, but please clarify this detail also here so that the reader will not be misled or feel the need to check the detail in the methods.

L.292-293: Although exclusion of the *Henriciella* data apparently did not alter the outcome of the clustering analysis, one should not simply "create" data for it. Instead, this strain should be excluded from the clustering analysis, as there is insufficient data to include it. For example, "setting the data to zero" is questionable given that Cnet in the algal cells is significant when co-cultured with this strain, as shown in Fig. S4, suggesting that this strain did metabolize algal DOC. What if *Henriciella* behaved similar to *Pusillimonas* or *Devosia*? (I chose these strains because the algal Cnet and Nnet were similar to those observed with *Henriciella*.) Clearly, we do not want to use "what if" arguments.

L.325-326: the presentation "with 3% > 2-day Cnet >1.5%" is rather confusing. Please consider reformulating it as "with 2-day Cnet between 1.5-3.0%", and similar later on ("between 0-1%").

L.330: Similar as above. Additionally, use 2-day instead of 48 h for consistency.

L.333: This reference should be Fig. S7B, I believe.

L.330-340: Please clarify what controls were used in these experiments.

L.340: This seems to be untrue for *Roseibium*. In Fig. S4A, *Roseibium* is indicated with an asterisk for Nnet DON (and also for Cnet DOC).

L.364: If I read the figure correctly, *Oceanicaulis* should be included in the list after *Yoonia*. Please check carefully.

L.390: Not clear why "phylogenetic". "genetic" seems more logical.

L.410: What is meant by "sugar alcohols" here?

L.429-431, 434: In these sentences, it would seem appropriate to incorporate/note the evidence for their high ability to use nitrate as an N-source.

L.437: It seems more appropriate to use "conclusions" instead of "categorizations".

L.441-460: I thank the authors for including this thoughtful discussion paragraph on cell-to-cell variability. I further encourage them to reflect in their discussion on the findings in the following papers (doi.org/10.3389/fmicb.2021.621634, doi.org/10.3389/fmicb.2021.620915), which discuss additional sources of cell-to-cell variability observed in single-cell NanoSIMS data, and particularly

in data obtained via dual-label nanoSIP experiments (¹³C and ¹⁵N), including storage compounds, cell cycle, or the ability to utilize C and N from multiple sources.

L. 445-447: The logic behind the suggestion is not clear.

L.452: Culturing in chemostats does not necessarily imply homogeneity in isotope labeling among cells. Measured isotope labeling in a chemostat-grown clonal population can still be very heterogeneous among cells (doi.org/10.3389/fmicb.2021.620915). On the other hand, nanoSIP experiments performed with cells grown in batch mode do seem to lead to higher observed cell-to-cell variability in their isotope labeling, including the presence of cells with an "unexpected" metabolic behavior (doi.org/10.3389/fmicb.2021.620915).

L. 466-467: Yes, but the substantial extra variability (indicated by the low R-squared) should also be acknowledged in this statement here.

L. 467-470: First, this discussion depends on the meaning of the word "utilizer". I think it would be more appropriate to use here the terms "incorporator" and "remineralizer" (used earlier in the text), which are more specific than "utilizer". Second, given that the ratio between the "fraction algal C from DOC" and "fraction bacterial C from DOC" is very similar for *Pusillimonas* and *Marinobacter*, the statement made for *Pusillimonas* can equally be made for *Marinobacter*. The difference between the two is that *Marinobacter* assimilates algal DOC and transfers the remineralized DOC to the alga more slowly. However, the efficiency, as measured by the above ratio, is very similar.

L.488-489: This statement illustrates an interesting contrast between C and N identified in this study. It would be helpful to illustrate it also graphically by visualizing a similar correlation as shown in Fig. S5A, but for N instead of C.

L.490: It would seem logical to add "and" before "commonly".

L.511-513: Please consider removing "either" before "phylogeny" (the two options are not excluding each other) and adding "derived from genomic data" after "metabolism". Also, there should probably be a comma before "however".

L. 522: Perhaps change "if" to "whether".

Table 1: As mentioned earlier, please be clearer here how the chlorophyll fluorescence was used to evaluate the effect of the bacterial strains on the algal yield. Also, include data on cell size and estimated cellular C and N content as supplementary data, for completeness.

Figure 1B: it would be better if the x and y axes had the same scale (0% to 1% in steps of 0.2%).

Figure 3: It would be better to use colors with a higher contrast for the two strains. These shades of green and blue are quite difficult to distinguish (at least for me on a printed version).

Figure 4: In the caption to this conceptual figure, it would be useful to be more specific about the meaning of "remineralization" as "transfer of remineralized algal-derived DOM". Particularly, "no N remineralization" on L. 738 may be misleading. N remineralization may occur, but the products are not transferred to/incorporated by the alga. Also, it should be made clear that this conceptual diagram applies for *Phaeodactylum tricornutum* from this study. The reason is the same as why the title of the study needed to be changed.

Supplement:

p.1: "There algal enrichments..." Perhaps "The algal enrichments..."?

p.2: "atm %" is very confusing. Atm usually refers to "atmosphere" (as a unit for pressure), but here you want to write "atom %".

p.3: Why is it necessary to introduce yet another abbreviation "at. %"? Check for consistency.

p.4: It is unclear why you mention how you calculated "atom %" first, if you do not make use of these quantities in the subsequent text (e.g., to explain how Xnet were calculated). Also, it would be more precise to add "Fs" between "of the newly synthesized biomass" and "relative", and remove the part with "where Fs is ... spiked substrate". Or, if you do want to mention atom fractions, it might be useful to also include a reference to doi.org/10.3389/fmicb.2021.621634, where the relationships between Xnet and the atom fractions of the cells and substrates are discussed in detailed.

Figure S2: As mentioned earlier, it needs to be made clearer how the significant effect of the different strains on the algal yield were determined. Intuitively, data points suggest significant effects for all strains given the consistently greater fluorescence values after day 6.

Figure S3: Are you sure that the first image shows an SE image? To me, it looks much more like an ¹²C/¹⁴N image. One can typically see filter pores in the SE image.

Figure S4: Please improve the formatting of the graphs; especially indicate clearly the ticks on the upper x-axis. Please include the Cnet and Nnet values for all measured cells as a supplementary data available to the reader.

Figure S5: For completeness, it would be useful to include a graph similar to panel A, but for N from DON. In panel A, please change scaling and tick locations of the axes to more reasonable values (e.g., from 0 to 0.08 instead of 0.079, in steps of 0.02; from 0 to 0.01, in steps of 0.002). Similarly in panel C. In the caption: strictly speaking, the 3 categories are best separated when plotted in the PCA dimensions 1 and 2 shown in panel B. I think what you mean to say here is that "Using measured data, these 3 categories were best separated when ...". The point being that the PCA dimensions generally represent a linear combination of the measured data, and are therefore quite abstract.

Figure S7: Please revise this figure and caption carefully. Panel (B) is not included in the intended panel B (the middle one). Panels A and B seem to be swapped (judging from what is written in the caption). Solid and open markers are difficult to resolve. Bold genus names are only applicable to panels A and B. Spelling: "internals" should be "intervals".

Figure S8: Please clarify whether we see single cells or colonies. Also, include the size of the scale bar.

Reviewer #2 (Remarks to the Author):

The authors have sufficiently addressed all comments from my previous review.

Below we address reviewer comments **in bold**.

Reviewer #1 (Remarks to the Author):

I appreciate author's careful attention and thoughtful response to my comments and suggestions from the first review round. The manuscript is now clearer, and the conclusions now more accurately reflect the experimental data. Also the extra discussion points add value to the manuscript.

I have additional comments/suggestions that need to be addressed to complete the journey. They are mostly specific, and often minor, and I will write them below based on the line numbers in the updated manuscript (the version without tracked changes).

One major point that requires more attention concerns the observed cell-to-cell variability in the axenic culture of *P. tricornutum* incubated with the algal-derived DOM. Presently, this variability appears to be "brushed over" as a minor detail. My suggestion is that the authors explore and discuss alternative interpretations of this data, because it may represent an "inconvenient truth" that questions the overall interpretation of the data (or at least of the NanoSIMS data), namely that some algal cells seem to be able to do alone the same thing as most of the algal cells can do with the help of bacteria. I provide a more thorough discussion about this issue below, and it would be useful if the authors discussed whether or not (and why not) this could be an alternative interpretation of their data.

-We have added some discussion on this

Another major point involves the interpretation of the chlorophyll fluorescence data in the context of the impact of bacterial strains on the algal yield. More details below.

I encourage the authors to carefully revise the entire Supplement. It contains a lot of important information, but its presentation feels at times quite sloppy, especially the clarity of some figures and quality of some captions (see specific comments below). For example, the title of the supplement is still the old one.

-Thank you for pointing this out. We have gone through the Supplement more carefully.

Additionally, after having gone through the data again, it would be useful if the single-cell Cnet and Nnet data (essentially all data presented in Figures S4 and S7) were made available through a public data repository or as supplementary data (i.e., not just figures) accompanying the manuscript. This is a common practice when publishing NanoSIMS results these days, allowing others to analyze the data from other perspectives, should they want to do that, and facilitating complete transparency of the publishing process.

-Thank you for this suggestion. We have made an excel file of the Cnet and Nnet data from the organic matter and glycolate and nitrate additions and provide this as a supplementary file.

Specific comments:

L.23: Strictly speaking “bacterial remineralization” cannot be analyzed at a single-cell level with nanoSIMS, as only *assimilated* C or N (by bacteria or the algae) can be quantified. Indeed, for the remineralization part, what is measured and interpreted is the remineralized algal-derived DOM transferred to, and incorporated by, the alga. It would be good to reformulate this sentence accordingly. I tried but could not find a “quick fix”. I understand that the meaning of “remineralization” is clearly defined by the authors on L.87-90, but it should also be clear and precise in the abstract.

Amended to “we quantified bacterial incorporation of algal-derived complex dissolved organic carbon and nitrogen and algal incorporation of remineralized carbon and nitrogen in fifteen algal-bacterial co-cultures...”

L.30-31: This sentence is an important outcome of the study, but lacks a conclusive remarks that highlights the take-home message. It could be combined with what was written earlier, something along this idea: “... and could not be elucidated strictly from metabolic capacity as predicted by comparative genomics, highlighting the need for direct activity-based measurements in ecological studies of microbial metabolic interactions.”

Amended

L.102: Better write “our” instead of “these”, because in the preceding sentences you are not referring to any data as such, but rather to methodological approaches used to acquire them.

amended

L.143: Please use something like “regime” after “light/dark” and “prepared” before “using Instant Ocean salts” in this sentence. Also, try to revise this very long and complex sentence, possibly by cutting it into two simpler, more straight-forward sentences.

amended

L.145: “We tested growth impacts” – yes, but please clarify for which purpose and what specifically was used as a measure of “growth”. This was an issue earlier, and I think it has not been resolved satisfactorily yet, as I try to explain in the following.

We have amended this section, including renaming the subheading to “Cultivation conditions and quantification of growth”. We used *in-vivo* fluorescence as a proxy for algal biomass.

It is explained now that chlorophyll fluorescence was measured and used as a measure of algal biomass. It is also explained that the fluorescence at the last time-point (18 day) was used to test the impact on growth. But this latter choice seems to be rather arbitrary. Why was it not tested for example at day 15, or 13? It would seem more appropriate to use the *entire* time-series (shown in Fig. S2) and explain why the variations of algal fluorescence over time are significantly affected by the bacteria for some of the alga-bacteria co-cultures, and not for some others, although they all show pretty much the same shape (as far as I can see in panel D in comparison to panels A-C). Specifically, the patterns in Fig. S2 seem to group into two groups based on the difference on day 4. From day 7-8 onward, the mean values appear to be systematically higher (although probably not always significantly higher for a given individual time point) for all co-cultures compared to the axenic culture. Yet, the conclusion is that only for 3 out of the 15 co-cultures this effect is significant, which I find quite counter-intuitive.

Thank you for this insight. Indeed, there appears to be a separation between 2 groups of co-cultures. For example, at day 4, one group has fluorescence values less than the axenic, and the other has fluorescence equal to the axenic. We have added this detail in the supplemental document. We focused on the later timepoints in these experiments rather than the early timepoints because we were primarily interested in understanding if the presence of specific bacterial strains led to higher algal biomass caused by increased bacterial remineralization of algal organic matter, and we do not expect this mechanism to occur early in the growth phase. We also note that the higher chlorophyll fluorescence measurements of *Marinobacter*, *Devosia*, and *Alcanivorax* co-cultures compared to axenic are in agreement with increased algal cell abundances previously published in Chorazyczewski et al. 2021. Also note that we have carried out a number of such growth experiments (not shown), and *Marinobacter*, *Devosia*, and *Alcanivorax* consistently show increased algal biomass as measured by in-vivo fluorescence, so the experiment shown on Figure S2 is representative of others. For the sake of simplification, we did not include those other experiments (done in 24 well plates under different light, nutrient, and temperature). If the reviewer wishes, we could include those additional experiments in the supplement.

Also, it can be assumed that the “growth impact” will depend on the number of bacterial cells in the co-culture and their activity. If the activity were the same for the different bacterial strains (which they were not, but suppose now that they were), the impact of a particular strain could be detectable if the corresponding cell counts were larger, whereas the impact could be undetectable if the cell counts were lower. In other words, cells counts of the bacteria in the co-cultures matter probably as much as their activity. But the bacterial cell counts (such as in the initial bacterial inoculum) were not under direct control in these experiments, or were they? If not, how can we conclude anything about the impact of a specific bacterial strain if we do not account for their abundance?

We appreciate this interesting discussion. Adding each bacterial strain at the same abundance and testing its influence on algal yield would be one experimental approach. The alternative, as described in the manuscript, was to establish the co-cultures with an initial bacterial inoculation and transferring the co-cultures for 6 months in order for each co-culture to be “in balance”, where the bacteria maintain their numbers only through their ability to grow in the presence of the algae. This, indeed, led to different numbers of bacterial cells (shown in Table 1), which is precisely what we were trying to achieve: bacteria better able to grow on algal exudates would be more abundant, and those less able to grow on the exudates would be less abundant. We have added a bit more info to the introduction and materials.

Overall, I am quite confused as to the purpose of the chlorophyll fluorescence assays, and how their results, alone and in combination with the results of the nanoSIP experiments, help us understand the role (impact) of the different bacterial strains.

We hope the response above clarified this confusion (pasted here again): “We focused on the later timepoints in these experiments rather than the early timepoints because we were primarily interested in understanding if the presence of specific bacterial strains led to higher algal biomass caused by increased bacterial remineralization of algal organic

matter, and we do not expect this mechanism to occur early in the growth phase.” We expected that bacteria that remineralize the most nutrients would be associated with increased algal biomass, and our data did not show this relationship. For example, the highest algal biomass were found in *Marinobacter*, *Devosia*, and *Alcanivorax* co-cultures which were on the lower spectrum of remineralized N to the algae.

I think the confusion stems from the unclear description in the “Algal growth effects and microscopy” section. What is meant by “any other experiments” on L. 137?

This meant to say “all experiments”, this section has been amended and we hope it is more clear.

Did you grow the co-cultures for >6 months (>6 transfers, monthly) and then used a subsample from them to conduct the nanoSIP incubation experiments? Similarly, did you use a similar subsample to inoculate 96 well plates with the F/2 medium, and then measured growth of the algae in these plates through monitoring of chlorophyll fluorescence? If this was the case, how was it ensured that the starting point of the co-cultures (algal cell count?) were the same in order to ensure that the time-series (shown in Fig. S2) could be comparable among each other?

Please explain these steps better, otherwise the origin of the co-cultures in each experiment, and the contribution of the “growth-impact” and nanoSIP experiments to the overall understanding of the diatom-bacteria interaction, are not clear. The reader should not be left guessing.

We have attempted to further clarify. The co-cultures were allowed to equilibrate (to be “in-balance”) for 6 months before any experiment was carried out, and all experiments were started at the same growth phase by using chlorophyll fluorescence to determine the inoculation volume. Note that the nanoSIMS, chlorophyll fluorescence growth impacts, and proteomics experiments were separate experiments.

L.155: “was added”. As explained in my previous comment, please better clarify the origin and history of the co-cultures just prior to the labeled substrate addition.

Amended here to “in-balance co-cultures”, hopefully, with the rewritten section above, this is now clear.

L.159: What do the “18” and “45” values mean in atom %? Is this a typo?

Rephrased to 18% ¹³C and 45% ¹⁵N

L.160: Is Ref. 29 (Dekas et al. 2019) is used to point the reader to the approach used for calculating Cnet and Nnet, or to clarify how to measure ¹³C and ¹⁵N enrichment of the exudate by nanoSIMS after spotting on a Si wafer?

-sorry for the confusion, it is the former and we have clarified

L. 165-169: I find this formulation somewhat confusing, hard to figure out the connection between the 11 strains and the 4 strains in the added sentence. Perhaps it would be clearer if written something like this: “For a third isotope tracing experiment, we tested the anapleurotic activity under extremely low nutrient conditions by incubating the strains in autotrophic medium (artificial seawater ESAW) with no added carbon plus ¹³C-labeled bicarbonate (1.5 mM added)

for one week. Results of these experiments are only available for 11 of the 15 strains because 4 strains showed signs of contamination.”

amended

L.171: It would be useful to start a new paragraph with “To compare across strains, …”

amended

L. 175-180: With this text, do you mean that the ¹³C and ¹⁵N atom fraction of the substrate taken up by the cells was assumed and not directly measured? Was it the same in incubations for all co-cultures? For the sake of transparency, it would be better to include more details on this in the supplement.

We have attempted to clarify this description and added additional text in the supplement.

L.192: It may not matter too much, but it would perhaps make more sense to define CD as a difference between Q1 and Q3 divided by the *average* of Q1 and Q3, so $CD = 2*(Q3 - Q1)/(Q3 + Q1)$.

We have renamed this the “quartile coefficient of dispersion”, which is defined by the original equation and we cannot change it: $(Q3 - Q1)/(Q3 + Q1)$

L.192-195: Here, you extrapolate from single-cell Cnet and Nnet data to the entire bacterial population. It is unclear, however, why you do this.

We made this calculation based on this reviewer’s comment from the previous revision asking for calculations of cell-specific rates of C and N incorporation. We did that but went a step further and incorporated cell abundances to calculate C and N incorporation per volume. This calculation shows how cell size, and more importantly, cell abundances impact the total amount of C and N incorporated by a bacterial population.

Additionally, the population-specific C and N assimilation depends on Cnet and Nnet of individual cells, cell counts, and on the cell size/biovolume (and thus the cellular C and N content). These latter data are, however, not provided, making it impossible to qualify whether the differences in the total DOC and DON daily incorporation among strains (reported in Table 1) are due to differences in Cnet and Nnet, cell counts, or cell size/biovolume. The cell size, biovolume, and C+N content data could be included as an “extended” version of Table 1 in the Supplement along with the Cnet+Nnet values and cell count data.

We have added an extended version of Table 1 to the Supplemental Material

Last but not least, when describing the process of estimating the total DOC and DON daily incorporation, it would be good to reflect on the findings in this paper, doi.org/10.3389/fmicb.2021.621634 (“Calculation and Interpretation of Substrate Assimilation Rates in Microbial Cells Based on Isotopic Composition Data Obtained by nanoSIMS”), which discusses in quite some detail how to extrapolate single-cell isotope enrichment data to population-specific C or N assimilation rates.

Thank you, we have added this reference, and a statement that our calculations are likely to be underestimates based on the assumption of linear isotope incorporation over the incubation time.

L. 219: “maximum *P. tricornutum* chlorophyll yield”. This is contradicting what you describe in the methods section as a “surrogate for growth”. In methods, you write that you measured chlorophyll fluorescence. I assumed that it is the fluorescence yield measured at very low (actinic) light in dark-adapted cells, which is an accepted way to approximate algal biomass. There is also *variable* chlorophyll fluorescence, which is used to quantify a quantum yield of PSII, which is a proxy for photosynthetic electron transport. In your expression here, you mix “maximum” (which maximum?) with chlorophyll yield (fluorescence or quantum yield?), and additionally use “algal yield” (in Table 1) or just “yield” (in Fig. S2). Together, these ambiguities make everything rather confusing. Please rewrite it more accurately and check for consistency between the main text and supplement.

We apologize for the confusion caused by the language. We did not measure fluorescence quantum yield (from dark-adapted cells), but more simply in-vivo chlorophyll fluorescence, also an accepted way to approximate algal biomass with live samples; see the first ever reference here <https://academic.oup.com/icesjms/article/30/1/3/642386> and a recent one here <https://www.ncbi.nlm.nih.gov/pmc/articles/PMC9368473/>. We have a regression curve showing statistically significant agreement between in vivo fluorescence and *P. tricornutum* cell counts, we can add this information to the supplementary if the reviewer wishes. We hope that our edits now make our approach more clear.

L.254: “highest incorporator”. As noted above, to which extent is this because of the highest cell counts?

Indeed, it is primarily due to high cell counts (see extended Table 1). We have amended this to point this out.

Also, for consistency of notation, it would be better to change in Table 1 the units from microgram DOC and microgram DON to microgram C and microgram N.

Label changed to microgram C from DOC and micrograms N from DON

L.257: again, it would be useful to clarify to which extent this is due to the high cell counts.

These two isolates had the highest cell abundances in co-culture, so this is mostly due to cell counts. Amended.

L.260: “...as well as at the single cell level ($p < 0.001$)”. With this add-on, the sentence as a whole does not read well. Please revise.

Sentence has been altered.

Also, it is unclear whether this correlation at the single cell level was observed for each bacterial strain separately or when Cnet & Nnet values from all bacterial cells were tested together. I assume the authors report the result for the latter test, but it would also be useful to test the correlations separately for each individual strain to quantify the coupling between the uptake of DOC and DON in the individual bacterial strains (strong and weak coupling would be indicated by high and low R-squared, respectively). I appreciate that this is not the focus of this study, but on the other hand, the data is there, so it's only a simple step to visualize it in a graph (or multiple graphs, if one graph gets over-crowded) and report R-squared and p-values. This information could be included in a Supplementary Figure and Table, and the corresponding text (probably 1 sentence) with the key outcome could be included at the end of the paragraph (L.

266+). I ask for this because it becomes relevant in the discussion about the sources of cell-to-cell variability in Cnet and Nnet.

We have now added these data in the supplement. Indeed, some strains had well correlated Cnet and Nnet, and others did not, suggesting coupling and uncoupling of DOC and DON incorporation, respectively.

L. 266: For completeness, it would be useful to mention that Yoonia and Pusillimonas lie close to the stoichiometric line, as opposed to above or below that line.

Amended.

L. 268: At this point of reading, it is difficult to recall the experimental design (three separate SIP experiments). Therefore, it would be useful to include a short clause at the beginning of this sentence to indicate, and in fact emphasize, that this algal data (algal Cnet and Nnet) is from the same SIP experiment (incubations) as the bacterial Cnet and Nnet data reported in the preceding paragraph.

Amended

L. 270-273: This finding deserves more attention. First, for the sake of completeness, please clarify what you mean by “minority”. What percentage of the total number of cells belong to this sub-population?

Added the percentage (18%)

Second, although cells in this sub-population may be a minority, their ^{13}C enrichment (Cnet) is pretty much in the same range (about 0.4 to 1.2 %) as the ^{13}C enrichment in algal cells co-cultured with bacteria. This is interesting and can be interpreted in different ways, as I try to outline below.

One possible interpretation is that a fraction of the algal DOC has a form (a specific form, albeit yet unknown) that can readily be re-assimilated by the algae without a need for modification (“remineralization”) by bacteria. That only a small fraction of algal cells does this is intriguing and should have a reason. Although I cannot think of one right now, the observation of that sub-population clearly points to significant differences among (presumably genetically equal) algal cells. Perhaps the studied algal species has a “special” metabolic potential to readily assimilate certain forms of DOC, and some of the cells do express this potential randomly, e.g., as a result of stochastic gene expression, during a 24 hr period.

Considering this “background” interaction (axenic alga vs. algal-DOC), the role of bacteria in the co-cultures must then be considered *relative* to this “background”. One possibility is that the bacteria modify (“remineralize”) the algal DOC such that the remineralized products can be assimilated by the *majority* (essentially all, as seen for most of the co-cultures) of the algal cells, i.e., not only by those that expressed a “special” ability, as suggested above. Intriguingly, the incorporation rate of the “virgin” DOC is very similar to that of the bacteria-remineralized DOC (as far as I can see in Fig. S4-B). Hence, on a single-cell level, it appears that the bacteria do *not* to speed-up the transfer of alga-derived DOC to algal cells, but rather make the alga-derived DOC available to more (all) algal cells. This rises a possibility of another interpretation: that the bacteria do not mineralize the DOC at all, but rather provide a signal to the algal cells that there is plenty of DOC available, which triggers *all* algal cells (as opposed to only those few that engaged in DOC incorporation “randomly” as a result of stochastic gene expression, as

suggested above) to switch to a metabolism involving incorporation of the “virgin” DOC. Although this latter possibility seems somewhat stretched, it is consistent with the NanoSIMS data presented in Fig. S4-B.

My point here is that the authors should explore alternative interpretations of their data, such as the one proposed above based on the NanoSIMS data available during this review process. The authors may incorporate the above hypothesis, or refute it if they have the required evidence. In any case, the minimum that the authors need to do is to revise their conclusion that “a minority of the [algal] cells potentially incorporated DOC”. It is not true that the cells “potentially” incorporated DOC. Their NanoSIMS data show that those algal cells *did* incorporate DOC (as seen by the distinct ^{13}C enrichment of the cells), and they did so rather significantly. It means something, but presently it is presented as a minor detail. I think this detail deserves attention and proper discussion, because it may represent an “inconvenient truth” that questions the overall interpretation of the data (or at least of the NanoSIMS data alone), namely that some algal cells seem to be able to do alone the same thing as most of the algal cells can do with the help of bacteria.

Thank you for this extended discussion. We have slightly rephrased this section of the results, and discuss these ideas in the discussion

L.280: I think it would be more accurate to write “incorporation” instead of “metabolism”.
amended

L.281: “led to”. I think this word suggests a causal relationship, but such a relationship is not supported by the data. A correct formulation would be “was associated with”. Also, it would be useful to state that although the relationship was very close to significant, the predicting power of that relationship was very poor (as indicated by the low value of R-squared).
amended

L.282-286: In this sentence, it is unclear that the subject are the algal cells in co-cultures. I agree that the context is clear because the entire text is about co-cultures, but here it would be useful to re-emphasize it by writing something like this: “Regarding N, algal cells in nine co-cultures exhibited significant ^{15}N incorporation compared to the axenic culture, except when cultured with *Algoriphagus*, ...”

We have changed this sentence to remind the reader we are talking about the algal cells (note that the suggested edit changed the meaning of the sentence to say the opposite of what we meant to say).

L.290: “method identified three functional guilds”. Was this identified “by itself” or because max of 3 groups were allowed during the cluster analysis? Maybe it is mentioned in the methods, but please clarify this detail also here so that the reader will not be misled or feel the need to check the detail in the methods.

We used the elbow method to identify the optimal number of clusters, and have added this information here.

L.292-293: Although exclusion of the *Henriciella* data apparently did not alter the outcome of the clustering analysis, one should not simply “create” data for it. Instead, this strain should be

excluded from the clustering analysis, as there is insufficient data to include it. For example, “setting the data to zero” is questionable given that Cnet in the algal cells is significant when co-cultured with this strain, as shown in Fig. S4, suggesting that this strain did metabolize algal DOC. What if Henriciella behaved similar to Pusillimonas or Devosia? (I chose these strains because the algal Cnet and Nnet were similar to those observed with Henriciella.) Clearly, we do not want to use “what if” arguments.

We have removed Henriciella from the Kmeans clustering and from conceptual figure 4, and merged this short section with the previous paragraph.

L.325-326: the presentation “with 3% > 2-day Cnet >1.5%” is rather confusing. Please consider reformulating it as “with 2-day Cnet between 1.5-3.0%”, and similar later on (“between 0-1%”).
amended

L.330: Similar as above. Additionally, use 2-day instead of 48 h for consistency.
amended

L.333: This reference should be Fig. S7B, I believe.
Now Fig S8B.

L.330-340: Please clarify what controls were used in these experiments.
Amended (killed controls)

L.340: This seems to be untrue for Roseibium. In Fig. S4A, Roseibium is indicated with an asterisk for Nnet DON (and also for Cnet DOC).
Rephrased to “little or no DON”, since indeed, Roseibium incorporated significant (but little) DON. The Cnet DOC incorporation here is not relevant since the discussion is about N.

L.364: If I read the figure correctly, Oceanicaulis should be included in the list after Yoonia. Please check carefully.
Indeed, thank you for catching that error

L.390: Not clear why “phylogenetic”. “genetic” seems more logical.
We have rephrased it to genetic. The two are related, because there is a phylogenetic signal when we examine the presence of metabolic pathways, but we agree it is confusing to bring it up here.

L.410: What is meant by “sugar alcohols” here?
Amended (hydroxyl-containing carbohydrates)

L.429-431, 434: In these sentences, it would seem appropriate to incorporate/note the evidence for their high ability to use nitrate as an N-source.
N incorporation from nitrate, while statistically significant, was quite low (see Fig S8), so we are cautious about pointing this out here, but we stated “some N from nitrate” in both of these cases).

L.437: It seems more appropriate to use “conclusions” instead of “categorizations”.
amended

L.441-460: I thank the authors for including this thoughtful discussion paragraph on cell-to-cell variability. I further encourage them to reflect in their discussion on the findings in the following papers (doi.org/10.3389/fmicb.2021.621634, doi.org/10.3389/fmicb.2021.620915), which discuss additional sources of cell-to-cell variability observed in single-cell NanoSIMS data, and particularly in data obtained via dual-label nanoSIP experiments (¹³C and ¹⁵N), including storage compounds, cell cycle, or the ability to utilize C and N from multiple sources.

We have added one additional reference on the cyanobacterial C and N incorporation and a short discussion here, keeping it brief as the manuscript is already quite lengthy.

L. 445-447: The logic behind the suggestion is not clear.

We tried to clarify the link between heterogeneity of incorporation and remineralization and how that would benefit a bacterial population dependent on algal exudation. The logic here is that we hypothesize that higher cell-to-cell variability in uptake of a complex substrate corresponds to some cells being less efficient at incorporation which would leave more material for uptake by the algal cells. One potential mechanism might be that only some of the bacterial cells produce extracellular enzymes for the breakdown of the complex DOM, and other cells benefit without producing the enzymes, and the algal cells also accidentally benefit.

L.452: Culturing in chemostats does not necessarily imply homogeneity in isotope labeling among cells. Measured isotope labeling in a chemostat-grown clonal population can still be very heterogeneous among cells (doi.org/10.3389/fmicb.2021.620915). On the other hand, nanoSIP experiments performed with cells grown in batch mode do seem to lead to higher observed cell-to-cell variability in their isotope labeling, including the presence of cells with an “unexpected” metabolic behavior (doi.org/10.3389/fmicb.2021.620915).

We have amended this text to reflect this information

L. 466-467: Yes, but the substantial extra variability (indicated by the low R-squared) should also be acknowledged in this statement here.

Acknowledged (added “not strong”)

L. 467-470: First, this discussion depends on the meaning of the word “utilizer”. I think it would be more appropriate to use here the terms “incorporator” and “remineralizer” (used earlier in the text), which are more specific than “utilizer”. Second, given that the ratio between the “fraction algal C from DOC” and “fraction bacterial C from DOC” is very similar for *Pusillimonas* and *Marinobacter*, the statement made for *Pusillimonas* can equally be made for *Marinobacter*. The difference between the two is that *Marinobacter* assimilates algal DOC and transfers the remineralized DOC to the alga more slowly. However, the efficiency, as measured by the above ratio, is very similar.

Amended, indeed *Marinobacter* looks like *Pusillimonas*.

L.488-489: This statement illustrates an interesting contrast between C and N identified in this

study. It would be helpful to illustrate it also graphically by visualizing a similar correlation as shown in Fig. S5A, but for N instead of C.

Added as supplemental figure 6D

L.490: It would seem logical to add “and” before “commonly”.

amended

L.511-513: Please consider removing “either” before “phylogeny” (the two options are not excluding each other) and adding “derived from genomic data” after “metabolism”. Also, there should probably be a comma before “however”.

amended

L. 522: Perhaps change “if” to “whether”.

amended

Table 1: As mentioned earlier, please be clearer here how the chlorophyll fluorescence was used to evaluate the effect of the bacterial strains on the algal yield. Also, include data on cell size and estimated cellular C and N content as supplementary data, for completeness.

Amended with a brief change in the caption, and we have added size and C and N content to the supplementary table

Figure 1B: it would be better if the x and y axes had the same scale (0% to 1% in steps of 0.2%).

Amended

Figure 3: It would be better to use colors with a higher contrast for the two strains. These shades of green and blue are quite difficult to distinguish (at least for me on a printed version).

We have made the bars solid instead of hollow and slightly adjusted the green tone, we hope this works.

Figure 4: In the caption to this conceptual figure, it would be useful to be more specific about the meaning of “remineralization” as “transfer of remineralized algal-derived DOM”. Particularly, “no N remineralization” on L. 738 may be misleading. N remineralization may occur, but the products are not transferred to/incorporated by the alga. Also, it should be made clear that this conceptual diagram applies for *Phaeodactylum tricornutum* from this study. The reason is the same as why the title of the study needed to be changed.

Amended (figure caption changed)

Supplement:

p.1: “There algal enrichments...” Perhaps “The algal enrichments...”?

Amended to “these”

p.2: “atm %” is very confusing. Atm usually refers to “atmosphere” (as a unit for pressure), but here you want to write “atom %”.

amended

p.3: Why is it necessary to introduce yet another abbreviation “at. %”? Check for consistency.

Amended

p.4: It is unclear why you mention how you calculated “atom %” first, if you do not make use of these quantities in the subsequent text (e.g., to explain how Xnet were calculated). Also, it would be more precise to add “Fs” between “of the newly synthesized biomass” and “relative”, and remove the part with “where Fs is ... spiked substrate”. Or, if you do want to mention atom fractions, it might be useful to also include a reference to doi.org/10.3389/fmicb.2021.621634, where the relationships between Xnet and the atom fractions of the cells and substrates are discussed in detailed.

This section has been amended , we switched atom % to isotope fractions, and reference the equation from Legin et al 2014 that uses fractions to calculate Xnet.

Figure S2: As mentioned earlier, it needs to be made clearer how the significant effect of the different strains on the algal yield were determined. Intuitively, data points suggest significant effects for all strains given the consistently greater fluorescence values after day 6.

We hope the edits in the main manuscript are satisfactory and we have edited the caption for this figure to reflect them

Figure S3: Are you sure that the first image shows an SE image? To me, it looks much more like an 12C14N image. One can typically see filter pores in the SE image.

Indeed, the reviewer is correct, this was the 12C 14N image. We have added the SE image to this figure.

Figure S4: Please improve the formatting of the graphs; especially indicate clearly the ticks on the upper x-axis. Please include the Cnet and Nnet values for all measured cells as a supplementary data available to the reader.

Tick marks added and data file included in the supplement

Figure S5: For completeness, it would be useful to include a graph similar to panel A, but for N from DON. In panel A, please change scaling and tick locations of the axes to more reasonable values (e.g., from 0 to 0.08 instead of 0.079, in steps of 0.02; from 0 to 0.01, in steps of 0.002). Similarly in panel C. In the caption: strictly speaking, the 3 categories are best separated when plotted in the PCA dimensions 1 and 2 shown in panel B. I think what you mean to say here is that “Using measured data, these 3 categories were best separated when ...”. The point being that the PCA dimensions generally represent a linear combination of the measured data, and are therefore quite abstract.

Figure S6D added, scales adjusted.

Figure S7: Please revise this figure and caption carefully. Panel (B) is not included in the intended panel B (the middle one). Panels A and B seem to be swapped (judging from what is written in the caption). Solid and open markers are difficult to resolve. Bold genus names are only applicable to panels A and B. Spelling: “internals” should be “intervals”.

Figure revised. We have made markers all the same since the asterisk indicates which are statistically significantly different.

Figure S8: Please clarify whether we see single cells or colonies. Also, include the size of the scale bar.

Amended